# Towards Faithful Reasoning in Remote Sensing: A Perceptually-Grounded GeoSpatial Chain-of-Thought for Vision-Language Models

**Jiaqi Liu**[1,2*], **Lang Sun**[1,2*], **Ronghao Fu**[1,2†], **Bo Yang**[1,2†]
[1]College of Computer Science and Technology, Jilin University, Changchun 130012, China
[2]Key Laboratory of Symbolic Computation and Knowledge Engineering of Ministry of Education
`{liujq21,sunlang24}@mails.jlu.edu.cn, {furh,ybo}@jlu.edu.cn`

## ABSTRACT

Vision-Language Models (VLMs) in remote sensing often fail at complex analytical tasks, a limitation stemming from their end-to-end training paradigm that bypasses crucial reasoning steps and leads to unverifiable outputs. To address this limitation, we introduce the Perceptually-Grounded Geospatial Chain-of-Thought (Geo-CoT), a framework that models remote sensing analysis as a verifiable, multi-step process. We instill this analytical process through a two-stage alignment strategy, leveraging Geo-CoT380k, the first large-scale dataset of structured Geo-CoT rationales. This strategy first employs supervised fine-tuning (SFT) to instill the foundational cognitive architecture, then leverages Group Reward Policy Optimization (GRPO) to refine the model's reasoning policy towards factual correctness. The resulting model, RSThinker, outputs both a final answer and its justifying, verifiable analytical trace. This capability yields dominant performance, significantly outperforming state-of-the-art models across a comprehensive range of tasks. The public release of our Geo-CoT380k dataset and RSThinker model upon publication serves as a concrete pathway from opaque perception towards structured, verifiable reasoning for Earth Observation.

## 1 INTRODUCTION

Vision-Language Models (VLMs) are rapidly redefining the analytical landscape for remote sensing, offering unprecedented capabilities for interpreting Earth Observation data (Kuckreja et al., 2024; Zhang et al., 2024; Soni et al., 2025; Pang et al., 2025; Fu et al., 2024b;a). These capabilities are demonstrated across a diverse array of downstream tasks, from complex visual question answering (VQA) to fine-grained object counting. Yet, the prevailing paradigm of these models involves learning an implicit, end-to-end mapping directly from pixels to a final output. Such an implicit mapping, by collapsing the entire reasoning process into a monolithic transformation, lacks procedural transparency and is consequently prone to generating plausible yet factually ungrounded hallucinations. The risk of such hallucinations presents a formidable barrier in high-stakes remote sensing applications, like disaster response (Misra et al., 2025; Lenton et al., 2024) or environmental monitoring (Wang et al., 2025; Silsbe et al., 2025), where the verifiability of a result is paramount. In these critical applications, the ultimate utility of a model hinges not merely on the correctness of its output, but on the verifiability of the process that produced it.

This demand for a verifiable process motivates a paradigm shift from passive recognition to goal-directed active perception, a potential unlocked by the Multimodal Chain-of-Thought (MM-CoT) paradigm (Mitra et al., 2024; Shao et al., 2024; Gao et al., 2025). The promise of MM-CoT lies in its capacity to formulate and externalize an analytical plan, thereby transforming a model from a black-box recognizer into a methodical analyst. The necessity for such an explicit plan is uniquely acute

---

*Equal contribution.
†Corresponding authors.

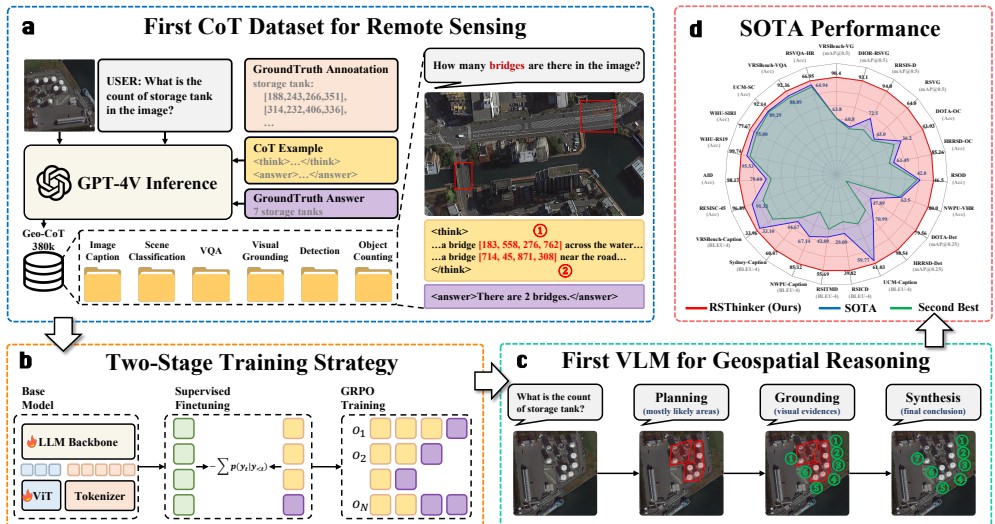

Figure 1: **An overview of the RSThinker framework.** Our novel Geo-CoT380k dataset (a) enables our two-stage alignment strategy (b) to instill a verifiable reasoning process (c), yielding state-of-the-art performance across a comprehensive suite of benchmarks (d).

in Earth Observation, where analytical challenges are multifaceted and deeply intertwined. This complexity directly translates into the challenge of navigating the sheer scale of regional-scale imagery with systematic search strategies, a requirement exemplified by tasks such as object counting. These strategies must in turn be guided by a forensic discrimination of subtle textural cues to resolve semantic ambiguities. This entire analytical process is often further constrained by the prevalence of topologically-grounded queries, demanding computational paths such as tracing a river network to locate every crossing bridge. These expert strategies, when externalized into a structured and verifiable sequence, constitute what we introduce as the Geospatial Chain-of-Thought (Geo-CoT).

Despite the clear need for such a Geo-CoT, prevailing approaches often frame reasoning as a process of semantic interpretation rather than visual investigation (Li et al., 2025a; Zhu et al., 2025). This interpretation relies on the model's parametric world knowledge for high-level deductions, such as identifying a stadium as a suitable evacuation point post-earthquake, rather than grounding its claims in immediate visual evidence. Even when contemporary models do attempt to incorporate visual evidence (Yao et al., 2025; Hu et al., 2025), it is typically presented as non-localizable text, mentioned without a verifiable link to a specific pixel region, thus leaving its claims unsubstantiated against hallucinated artifacts. This absence of a verifiable link stems from a more fundamental limitation: the lack of an intent-driven process for active perception. Instead of formulating and executing a decomposed analytical plan, these models perform a holistic, single-pass inference over the entire scene. This reactive inference is incapable of the systematic evidence gathering required for faithful reasoning, leaving a critical gap between the conceptual promise of MM-CoT and its practical realization in Earth Observation.

To bridge this critical gap in Earth Observation, we introduce a novel framework that instantiates the Perceptually-Grounded Geospatial Chain-of-Thought (Geo-CoT) within Vision-Language Models. Our framework materializes a rigorous cognitive architecture whose foundational principle is strict perceptual grounding, where abstract claims are replaced by assertions explicitly linked to specific spatial references. The operational flow of this grounding process follows a clear protocol of task planning, iterative evidence gathering, and final synthesis, enabling the VLMs to perform methodical visual interrogation rather than a reactive, holistic inference. We instill this reasoning protocol by first constructing Geo-CoT380k, a large-scale dataset populated via a scalable pipeline that retrofits verifiable rationales onto ground-truth data, and then leveraging this dataset in a two-stage alignment strategy. This strategy, a paradigm informed by recent large-scale LLM development (DeepSeek-AI, 2025; Guo et al., 2025), effectively decouples the architectural challenge of instilling a cognitive structure from the policy challenge of refining its factual correctness. Our first stage, supervised fine-tuning (SFT), establishes the foundational cognitive structure, followed by a subsequent stage leveraging Group Relative Policy Optimization (GRPO) to steer the model's generative process towards high-fidelity reasoning chains. Our primary contributions can be summarized as follows:

- We define and formalize the Perceptually-Grounded Geo-CoT, a reasoning paradigm for remote sensing that mandates a verifiable link between each analytical step and its corresponding visual evidence.

- We construct the first large-scale supervised fine-tuning (SFT) dataset for remote sensing chain-of-thought, Geo-CoT380k, explicitly designed to instill the cognitive architecture of task decomposition, iterative evidence grounding, and final synthesis.

- We present RSThinker, a VLM embodying our framework, demonstrating that a two-stage alignment strategy of SFT as a prerequisite for reinforcement learning (GRPO) is essential for faithfully eliciting this capability and setting a new state-of-the-art on a suite of canonical remote sensing tasks, including visual question answering and object counting.

## 2 RELATED WORK

### 2.1 VISION-LANGUAGE MODELS IN REMOTE SENSING

The application of Vision-Language Models (VLMs) to remote sensing has recently catalyzed a surge of innovation, fundamentally altering interactions with Earth Observation data. Pioneering works such as GeoChat (Kuckreja et al., 2024) and EarthGPT (Zhang et al., 2024) established the viability of equipping VLMs with the capacity for geospatial dialogue and handling a wide spectrum of queries. Subsequent models like EarthDial (Soni et al., 2025), VHM (Pang et al., 2025), SkyMoE (Liu et al., 2025b) and GeoDiT (Liu et al., 2025a) further refined this interactive paradigm through enhanced conversational fluency and novel architectural designs, achieving state-of-the-art performance on canonical benchmarks. Yet, a common architectural paradigm unites these powerful models: they are fundamentally optimized to map visual inputs to a final textual output. This end-to-end optimization, while successful, inherently treats the intermediate reasoning process as a latent and inaccessible variable. Consequently, a critical gap persists: the lack of a VLM capable of not only producing a correct answer, but also externalizing the verifiable, step-by-step analytical process that justifies it. Our work is explicitly designed to bridge this gap.

### 2.2 CHAIN-OF-THOUGHT AND REASONING IN VISION-LANGUAGE MODELS

The pursuit of a verifiable analytical process finds its intellectual origins in Chain-of-Thought (CoT) reasoning, a paradigm first established to elicit step-by-step thinking in language models. This paradigm has recently evolved into Grounded CoT within the general computer vision community, where abstract reasoning is explicitly anchored to visual evidence. Pioneering frameworks such as Visual CoT (Shao et al., 2024), VoCoT (Li et al., 2025b) and Argus (Man et al., 2025) have demonstrated the efficacy of interleaving bounding boxes within reasoning traces, while approaches like V* (Wu & Xie, 2024) and CMMCoT (Zhang et al., 2025) have explored guided visual search and memory augmentation to handle complex contexts. This methodological progression has demonstrated remarkable success in domains predicated on the presence of salient, well-defined entities. Existing frameworks thrive by reasoning over holistic objects, such as vehicles in traffic scenes (Wang et al., 2024; Mandalika et al., 2025) or instruments in medical images (Liu et al., 2024a; Jiang et al., 2025). However, this reliance on discrete, salient objects reveals a fundamental perceptual mismatch with the nature of Earth Observation. Remote sensing data is typically characterized by vast, non-uniform scenes and high-density, tiny objects that lack the semantic salience found in natural or medical photography. Consequently, generalist grounded models often falter in this domain, due to the lack of a domain-specific substrate, comprising large-scale specialized datasets and adapted cognitive architectures, necessary to render this concept operational and robust for Earth Observation.

### 2.3 REASONING IN REMOTE SENSING VISION-LANGUAGE MODELS

The pioneering efforts to apply reasoning chains within geospatial contexts have recently begun to emerge. In the broader geographic domain, frameworks like GeoChain (Yerramilli et al., 2025) and GAEA (Campos et al., 2025) have effectively utilized CoT for geolocation and landmark analysis. However, these approaches primarily address semantic reasoning in ground-level imagery, relying on cultural or architectural cues for knowledge retrieval. In the specific domain of overhead Earth

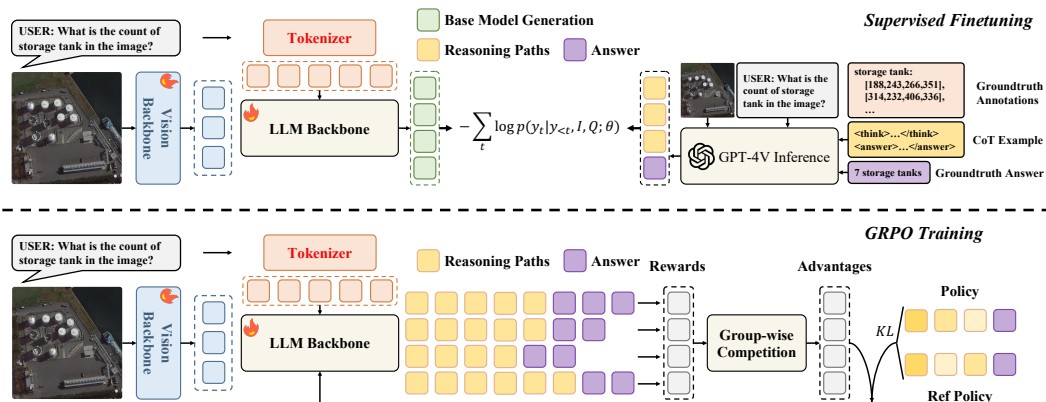

Figure 2: **The two-stage alignment process.** Our training strategy first instills a foundational cognitive architecture via supervised fine-tuning (SFT) and then refines this architecture's faithfulness via outcome-based reinforcement learning (GRPO).

Observation, works like SegEarth-R1 (Li et al., 2025a) and RemoteReasoner (Yao et al., 2025) have demonstrated the potential of generating step-by-step rationales to guide complex downstream tasks, while others such as SkySense-O (Zhu et al., 2025) have advanced the quality of these textual rationales. Even agentic frameworks like Ringmo-Agent (Hu et al., 2025) have emerged, capable of formulating high-level plans. However, a close examination reveals that these foundational frameworks share critical limitations. First, their reasoning steps often remain as abstract textual descriptions, lacking the direct, verifiable link to spatial areas that constitutes true perceptual grounding—a challenge uniquely acute in top-down views characterized by dense objects and scale variations. Second, the reasoning process itself, while sequential, typically lacks a methodical cognitive architecture. These explorations thus underscore a clear and unmet need for a framework that not only prompts for reasoning but fundamentally structures it around the principles of perceptual grounding and a systematic cognitive plan. Our work is the first to propose such a framework.

## 3 METHODOLOGY

To realize the Perceptually-Grounded Geospatial Chain-of-Thought (Geo-CoT) framework, we develop RSThinker, a foundational Vision-Language Model trained via a two-stage alignment process. This process is designed to instill the core cognitive architecture of Geo-CoT and subsequently refine its faithfulness. The initial stage of this process instills the foundational cognitive architecture of Geo-CoT, leveraging a large-scale supervised fine-tuning (SFT) corpus we constructed to explicitly embody the principles of task decomposition and iterative evidence grounding. The second stage subsequently employs reinforcement learning to refine the model's reasoning, guided by a domain-specific reward function we designed to optimize for the faithfulness of the grounded evidence. The resulting model, which we name RSThinker and illustrate in Figure 2, is thus a specialist VLM that reasons faithfully and remains verifiably grounded in visual evidence.

### 3.1 BASE VISION-LANGUAGE MODEL

We initialize RSThinker from the pre-training checkpoint of GLM-4.1V-9B-Base (Team et al., 2025b), a state-of-the-art VLM. Its architecture employs a Vision Transformer, Aimv2-Huge (Fini et al., 2025), which is particularly suited for remote sensing due to its ability to handle variable image resolutions and aspect ratios. This crucial capability is realized through a dynamic positional encoding scheme that adapts its pre-trained position table, $P_{orig}$. Specifically, the scheme first normalizes each patch coordinate $g = (w, h)$ to a continuous grid $g_{norm}$ spanning [-1, 1], and then samples from $P_{orig}$ via bicubic interpolation to compute the adapted encoding $P_{adapted}$:

$$g_{norm} = (w_{norm}, h_{norm}) = 2 \cdot \left(\frac{w + 0.5}{W_p}, \frac{h + 0.5}{H_p}\right) - 1,$$

$$P_{adapted}(g) = \mathcal{I}_{bicubic}(P_{orig}, g_{norm}), \tag{1}$$

Table 1: The overview of the dataset Geo-CoT380k.

| Tasks | Datasets | Samples |
|---|---|---|
| VQA | VRSBench-train-VQA | 85,813 |
| Image Captioning | VRSBench-train-cap | 20,264 |
| | FIT-RS-cap | 65,197 |
| Scene Classification | NWPU-RESISC45-train | 31,500 |
| | AID-train | 10,000 |
| Visual Grounding | DIOR-RSVG-train | 34,744 |
| | VRSBench-train-VG | 35,967 |
| Object Counting | DOTAv2-train | 25,769 |
| | HRRSD-train | 24,784 |
| Object Detection | DOTAv2-train | 25,769 |
| | HRRSD-train | 24,784 |

Table 2: Additional Dataset for RL.

| Tasks | Datasets | Samples |
|---|---|---|
| VQA | RSVQA-HR-train | 67,228 |
| Image Captioning | NWPU-Captions-train | 28,350 |
| | RSICD-train | 10,921 |
| | RSTMD-train | 4,291 |

Table 3: Task-specific reward functions.

| Task | Reward Design Details |
|---|---|
| VQA & Scene Classification | Reward = $1.0, 0.6, 0.0$ for correct, partially correct, others |
| Visual Grounding | Reward = IoU |
| Object Counting | Reward = $1.0 - \alpha \times \frac{\text{MAE}}{\max(|\text{Ans}|, |\text{GT}|)}$ |
| Object Detection | Reward = mAP@0.5 |
| Image Captioning | Reward = $\sum_{m \in M} w_m \cdot m$ |

$m \in \{\text{BLEU-4, METEOR, CIDEr, ROUGE-L}\}$

This robust visual encoding mechanism, complemented by a 3D-RoPE language decoder for enhanced spatial awareness, provides a powerful and flexible foundation upon which we build our domain-specific alignment.

## 3.2 STAGE I: INSTILLING COGNITIVE ARCHITECTURE VIA SUPERVISED FINE-TUNING

The efficacy of our SFT stage is contingent upon a large-scale corpus of structured rationales that embody the Geo-CoT principles. To this end, we developed a scalable annotation pipeline that leverages a powerful, general-purpose VLM, GPT-4V (OpenAI, 2023), to generate these rationales. Our pipeline empirically promotes faithfulness through strict conditioning: rather than tasking the VLM with open-ended reasoning, we provide it with verified bounding boxes, image captions, and chain-of-thought exemplars (detailed in Appendix A.7), minimizing the risk of hallucinated reasoning. This methodology allows us to produce a vast, high-fidelity SFT-CoT dataset, Geo-CoT380k, comprising 384,591 structured rationales sourced from diverse, publicly-available remote sensing benchmarks (detailed in Table 1), including large-scale imagery from sources like DOTAv2 that was tiled into 800×800 patches.

With this dataset established, the SFT stage compels the VLM to internalize the entire methodical workflow encoded in each structured output $o_i$. This workflow, represented as <think> ... </think><answer> ... </answer>, is learned through a standard auto-regressive objective that maximizes the log-likelihood of the target rationale:

$$\mathcal{L}_{\text{SFT}}(\theta) = - \sum_{t=1}^{|o_i|} \log p(o_{i,t}|o_{i,<t}, I, Q; \theta), \tag{2}$$

By optimizing this loss function, we are not simply fine-tuning for a task; we are fundamentally reshaping the model's internal reasoning process to explicitly model the decomposition, grounding, and synthesis steps of the Geo-CoT cognitive architecture.

## 3.3 STAGE II: REFINING FAITHFULNESS VIA GROUP RELATIVE POLICY OPTIMIZATION

While the SFT stage successfully instills the structural template of Geo-CoT, its token-level maximum likelihood objective can still assign high probability to rationales that are locally plausible but contain unfaithful links between evidence and claims. To address these sequence-level deficiencies, our second alignment stage employs Group Relative Policy Optimization (GRPO), an outcome-based reinforcement learning paradigm wherein the reward signal is derived solely from the final output of the reasoning trace. For each task, this reward function directly embodies its canonical evaluation metric (Table 3), ensuring our policy optimization is precisely aligned with established performance protocols.

The GRPO training process directly optimizes the generative policy $\pi_\theta$ using on-policy sampling, drawing inputs from a designated preference tuning corpus comprising the original, rationale-free instances from Geo-CoT380k, augmented with additional datasets detailed in Table 2. Given an input $(I, Q)$ drawn from the dataset $\mathcal{D}$, we first sample a group of $k$ outputs, $\{o_1, o_2, \cdots, o_k\}$. The raw reward scores for each, $\mathcal{R} = \{\mathcal{R}_1, \mathcal{R}_2, \cdots, \mathcal{R}_k\}$, are then normalized to yield a low-variance

Table 4: Comparison of RSThinker with existing generic and RS VLMs on Visual Grounding task.

| Method | VRSBench-VG | | | DIOR-RSVG | | | RRSIS-D (ZS) | | | RSVG (ZS) | | |
|---|---|---|---|---|---|---|---|---|---|---|---|---|
| | @0.5 | @0.75 | mIoU | @0.5 | @0.75 | mIoU | @0.5 | @0.75 | mIoU | @0.5 | @0.75 | mIoU |
| *Close-source Commercial Vision-Language Models* | | | | | | | | | | | | |
| Claude-sonnet-4 | 11.1 | 2.4 | 16.66 | 17.6 | 1.2 | 25.33 | 20.5 | 1.5 | 29.91 | 24.0 | 7.0 | 24.99 |
| Gemini-2.0-flash | 22.9 | 6.3 | 28.59 | 20.8 | 3.3 | 27.45 | 29.5 | 5.0 | 36.13 | 19.5 | 4.5 | 24.07 |
| ChatGPT-5 | 14.4 | 2.3 | 22.71 | 26.1 | 3.3 | 28.37 | 28.0 | 5.0 | 29.46 | 18.5 | 3.5 | 20.59 |
| *Open-source Vision-Language Models* | | | | | | | | | | | | |
| MiniGPT-v2 | 32.1 | 16.3 | 33.96 | 29.4 | 10.2 | 29.43 | 38.5 | 16.0 | 40.13 | 12.0 | 3.0 | 15.65 |
| Qwen2.5-VL | 45.2 | 20.6 | 42.45 | 36.3 | 15.9 | 34.34 | 0.5 | 0.0 | 5.17 | 1.0 | 0.0 | 7.24 |
| *Open-source Reasoning Vision-Language Models* | | | | | | | | | | | | |
| GLM-4.1V-Thinking | 63.8 | 47.0 | 60.69 | 59.6 | 43.7 | 57.41 | 63.5 | 47.5 | 61.84 | 43.0 | 30.5 | 42.27 |
| *Open-source Remote Sensing Vision-Language Models* | | | | | | | | | | | | |
| GeoChat | 56.3 | 24.6 | 53.50 | 31.4 | 11.0 | 34.99 | 10.0 | 0.5 | 20.35 | 5.5 | 0.5 | 12.55 |
| VHM | 33.9 | 10.0 | 34.91 | 55.9 | 35.5 | 49.90 | 64.0 | 37.5 | 55.20 | 2.5 | 0.0 | 5.80 |
| SkySenseGPT | 63.5 | 26.0 | 54.60 | 60.8 | 26.5 | 53.18 | 69.0 | 32.5 | 59.87 | 39.5 | 17.5 | 38.54 |
| EarthDial | 14.4 | 7.8 | 13.04 | 46.1 | 30.2 | 39.46 | 72.5 | 50.0 | 64.08 | 42.0 | 24.0 | 38.49 |
| **RSThinker** | **90.4** | **77.2** | **80.79** | **93.1** | **90.2** | **89.02** | **94.0** | **90.5** | **89.59** | **64.0** | **54.5** | **59.74** |

Table 5: Comparison of RSThinker with existing generic and RS VLMs on Object Counting task.

| Method | DOTAv2-val | | HRRSD | | RSOD (ZS) | | NWPU-VHR (ZS) | |
|---|---|---|---|---|---|---|---|---|
| | Acc ↑ | MAE ↓ | Acc ↑ | MAE ↓ | Acc ↑ | MAE ↓ | Acc ↑ | MAE ↓ |
| *Close-source Commercial Vision-Language Models* | | | | | | | | |
| Claude-sonnet-4 | 25.17 | 10.232 | 50.11 | 2.231 | 25.0 | 4.115 | 51.5 | 2.205 |
| Gemini-2.0-flash | 29.36 | 15.057 | 54.65 | 1.921 | 39.0 | 4.095 | 63.5 | 1.835 |
| ChatGPT-5 | 36.20 | 7.490 | 58.50 | 0.787 | 40.0 | 1.430 | 58.0 | 1.310 |
| *Open-source Vision-Language Models* | | | | | | | | |
| MiniGPT-v2 | 10.82 | 57.082 | 19.50 | 36.059 | 19.5 | 9.630 | 21.0 | 4.675 |
| Qwen2.5-VL | 33.77 | 9.733 | 57.82 | 0.846 | 42.0 | 1.370 | 58.0 | 1.170 |
| *Open-source Reasoning Vision-Language Models* | | | | | | | | |
| Kimi-VL-Thinking | 30.68 | 11.967 | 46.26 | 1.612 | 15.5 | 4.050 | 53.0 | 2.575 |
| GLM-4.1V-Thinking | 29.80 | 8.072 | 58.96 | 0.903 | 28.5 | 3.220 | 62.5 | 1.194 |
| *Open-source Remote Sensing Vision-Language Models* | | | | | | | | |
| VHM | 32.67 | 9.260 | 46.71 | 1.063 | 16.0 | 1.791 | 48.5 | 1.289 |
| SkySenseGPT | 33.11 | 7.199 | 58.73 | 1.070 | **51.5** | 3.079 | 49.5 | 1.835 |
| EarthDial | 32.23 | 8.422 | 61.45 | 0.871 | 41.0 | 1.642 | 52.5 | 1.323 |
| **RSThinker** | **43.93** | **2.728** | **85.26** | **0.242** | 46.5 | 1.130 | **80.0** | **0.465** |

Figure 3: Comparison of RSThinker with SOTA VLMs on Object Detection task.

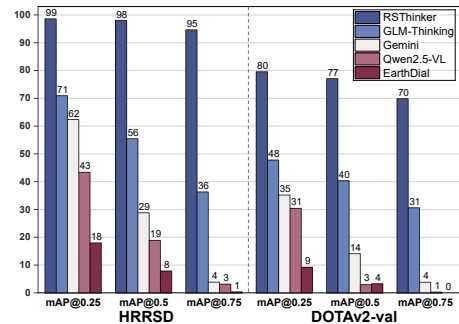

estimate of the group-relative advantage, $\hat{A}_i$. The policy is then updated by optimizing the following clipped surrogate objective:

$$\mathcal{L}_{\text{GRPO}}(\theta) = -\mathbb{E}_{\left[(I,Q)\sim\mathcal{D},\{o_i\}_{i=1}^k\sim\pi_{\theta_{old}}(\cdot|I,Q)\right]}$$

$$\sum_{i=1}^{k}\sum_{t=1}^{|o_i|}\min\left(r_{t,i}(\theta)\hat{A}_i, \text{clip}(r_{t,i}(\theta), 1-\epsilon, 1+\epsilon)\hat{A}_i\right) - \beta D_{\text{KL}}(\pi_\theta\|\pi_{\text{ref}}), \quad (3)$$

$$r_{t,i}(\theta) = \frac{\pi_\theta(o_{i,t}|q,o_{i,<t})}{\pi_{\theta_{old}}(o_{i,t}|q,o_{i,<t})}, \hat{A}_i = \frac{\mathcal{R}_i - \text{mean}(\mathcal{R})}{\text{std}(\mathcal{R})},$$

where the clip function constrains this ratio within the interval $[1-\epsilon, 1+\epsilon]$, thereby disincentivizing overly aggressive policy updates. The final term is a KL-divergence penalty that regularizes the policy $\pi_\theta$, preventing it from deviating excessively from the reference policy $\pi_{\text{ref}}$ (initialized from the SFT checkpoint). This optimization process systematically shifts the probability mass of the policy distribution, moving it away from regions that produce low-reward outcomes and towards those that generate high-reward, verifiably correct conclusions. This final alignment step imbues the model's internal reasoning process with a functional alignment to the ultimate goal of achieving factual correctness.

## 4 EXPERIMENT

We present a comprehensive experimental evaluation designed to validate our core contributions. This evaluation first establishes the state-of-the-art performance of our model, RSThinker, across a diverse suite of canonical remote sensing tasks. Beyond this aggregate performance, we conduct a series of carefully designed ablation studies to isolate the causal impact of each component of our framework. Finally, we provide a qualitative analysis to visually demonstrate the nature and faithfulness of the Perceptually-Grounded Geo-CoT that our framework uniquely produces.

Table 6: Comparison of RSThinker with generic and RS VLMs on Classification and VQA tasks.

| Method | Scene Classification | | | | | VRSBench-VQA | | | | | | | RSVQA-HR | |
|---|---|---|---|---|---|---|---|---|---|---|---|---|---|---|
| | RESISC45 | AID | RS19 (ZS) | SIRI (ZS) | UCM (ZS) | Category | Existence | Position | Quantity | Scene | Color | Image | Presence | Comp |
| *Close-source Commercial Vision-Language Models* | | | | | | | | | | | | | | |
| Claude-sonnet-4 | 58.44 | 60.33 | 76.32 | 64.33 | 67.86 | 43.28 | 52.78 | 30.17 | 66.67 | 64.79 | 63.29 | 91.67 | 46.95 | 64.94 |
| Gemini-2.0-flash | 74.89 | 76.00 | 90.00 | 72.00 | 85.95 | 44.03 | 86.11 | 43.97 | 46.00 | 60.56 | 56.96 | 95.83 | 56.94 | 42.96 |
| ChatGPT-5 | 82.22 | 75.50 | 95.53 | 75.00 | 88.57 | 39.55 | 88.89 | 42.24 | 47.33 | 70.42 | 59.49 | 87.50 | 62.94 | 68.93 |
| *Open-source Vision-Language Models* | | | | | | | | | | | | | | |
| MiniGPT-v2 | 32.67 | 27.17 | 30.79 | 26.67 | 32.86 | 25.37 | 56.25 | 20.69 | 44.00 | 45.07 | 36.71 | 33.33 | 48.95 | 52.95 |
| Qwen2.5-VL | 68.89 | 71.67 | 86.05 | 67.33 | 78.33 | 37.31 | 75.69 | 37.93 | 44.00 | 67.61 | 63.29 | 91.67 | 57.92 | 56.94 |
| *Open-source Reasoning Vision-Language Models* | | | | | | | | | | | | | | |
| Kimi-VL-Thinking | 72.22 | 70.50 | 88.68 | 69.00 | 77.62 | 47.01 | 87.50 | 46.55 | 74.67 | 71.83 | 65.82 | 90.23 | 63.94 | 77.91 |
| GLM-4.1V-Thinking | 70.09 | 69.67 | 86.84 | 60.33 | 82.86 | 42.54 | 86.11 | 43.10 | 54.67 | 69.01 | 62.03 | 87.50 | 45.95 | 65.93 |
| *Open-source Remote Sensing Vision-Language Models* | | | | | | | | | | | | | | |
| VHM | 91.33 | 79.00 | 91.84 | 64.33 | 89.29 | 50.75 | 86.81 | 36.21 | 42.67 | 53.52 | 55.70 | 54.17 | 61.94 | 76.92 |
| SkySenseGPT | 83.33 | 75.50 | 93.16 | 55.33 | 85.00 | 57.46 | 84.03 | 44.83 | 38.00 | 53.52 | 16.46 | 45.83 | 47.95 | 78.93 |
| EarthDial | 76.67 | 67.33 | 88.76 | 73.42 | 80.71 | 51.49 | 47.22 | 36.21 | 41.33 | 36.62 | 11.39 | 50.00 | 64.94 | 79.92 |
| **RSThinker** | **96.89** | **98.17** | **99.74** | **77.67** | **92.14** | **82.84** | **92.36** | **68.97** | 56.67 | **73.24** | 64.33 | 92.87 | **66.95** | 78.98 |

## 4.1 Experimental Setup

**Tasks and Benchmarks.** We validate the performance of RSThinker across a comprehensive suite of canonical remote sensing tasks. This evaluation spans the full spectrum from fine-grained, object-level analysis (object counting, detection, and grounding) to holistic scene interpretation and complex reasoning (classification, captioning, and VQA), with a detailed breakdown of all benchmarks provided in Appendix A.4.1.

**Baseline Models.** To contextualize RSThinker's performance, we conduct a rigorous comparison against a wide range of baseline models. These models are organized along two primary axes: their domain specialization (general-purpose vs. remote sensing) and their architectural support for explicit reasoning. This comparative analysis therefore includes leading proprietary systems, open-source generalist and domain-specific VLMs, and the latest reasoning-centric frameworks, a complete list of which is detailed in Appendix A.4.2.

**Implementation Details.** Our implementation of RSThinker is initialized from the GLM-4.1V-Base checkpoint, and its performance across all experiments is assessed using standard, community-accepted evaluation metrics. These metrics include mean Average Precision (mAP) and Intersection over Union (IoU) for object detection, Accuracy (Acc) and Intersection over Union (IoU) for visual grounding, Mean Absolute Error (MAE) for counting, Accuracy for classification and VQA, and BLEU-4, METEOR, and CIDEr for captioning. Further details regarding the full training protocol and hyperparameters are deferred to Appendix A.4.3.

## 4.2 Main Results and Analysis

We present a comprehensive evaluation of RSThinker against a suite of state-of-the-art models. Our analysis is structured around distinct categories of remote sensing capabilities, moving from fine-grained perception to holistic scene understanding and reasoning.

### 4.2.1 Fine-Grained Perception: Grounding, Detection, and Counting

The efficacy of the Geo-CoT framework is most directly validated in fine-grained perception, where the veracity of an output is inextricably linked to the model's ability to localize spatial evidence. This principle is clearly demonstrated in Visual Grounding (Table 4), a task demanding an explicit link between text and pixels. RSThinker establishes a substantial performance margin in this task, an advantage that stems from a fundamental architectural divergence. Baseline models typically rely on end-to-end architectures where grounding remains a latent, unconstrained variable within the network. In contrast, our two-stage alignment mandates that the model externalize and report specific, falsifiable spatial references, making a commitment to tangible evidence a required component of the output format.

This foundational capability for precise localization naturally extends to the more complex task of Object Detection (Figure 3). The Geo-CoT framework transforms detection from a single-pass recognition into a methodical, sequential search. Its Planning–Grounding–Synthesize structure compels a systematic scan of the imagery, a critical advantage that enables the exhaustive identification of objects in dense scenes where holistic approaches can fail. The benefits of this structured analytical process culminate in Object Counting (Table 5), which sees a significant reduction in Mean Absolute Error. This reduction in error is a direct consequence of the Geo-CoT architecture provid-

Table 7: Comparison of RSThinker with existing generic and RS VLMs on Image Captioning task.

| Method | RSITMD | | | NWPU-Captions | | | RSICD | | | VRSBench-Cap | | |
|---|---|---|---|---|---|---|---|---|---|---|---|---|
| | B-4 | MT | Cr | B-4 | MT | Cr | B-4 | MT | Cr | B-4 | MT | Cr |
| *Close-source Commercial Vision-Language Models* | | | | | | | | | | | | |
| Claude-sonnet-4 | 20.14 | 17.15 | 19.31 | 28.32 | 21.98 | 32.46 | 11.58 | 13.90 | 24.57 | 14.62 | 22.36 | 73.49 |
| Gemini-2.0-flash | 15.73 | 9.27 | 17.11 | 20.55 | 11.42 | 22.58 | 10.85 | 8.71 | 21.53 | 14.19 | 22.30 | 86.33 |
| ChatGPT-5 | 27.27 | 21.10 | 29.48 | 39.62 | 25.69 | 48.52 | 16.83 | 16.73 | 34.39 | 18.06 | **25.11** | 88.93 |
| *Open-source Vision-Language Models* | | | | | | | | | | | | |
| MiniGPT-v2 | 25.45 | 16.83 | 25.89 | 37.75 | 19.70 | 35.73 | 15.40 | 12.36 | 26.63 | 26.61 | 18.36 | 68.94 |
| Qwen2.5-VL | 27.92 | 17.24 | 24.90 | 38.89 | 21.40 | 42.11 | 17.80 | 13.72 | 32.19 | 29.21 | 25.01 | 91.84 |
| *Open-source Reasoning Vision-Language Models* | | | | | | | | | | | | |
| Kimi-VL-Thinking | 24.82 | 16.47 | 22.02 | 34.84 | 20.08 | 37.14 | 15.60 | 13.57 | 30.00 | 26.07 | 24.34 | 83.86 |
| GLM-4.1V-Thinking | 20.57 | 19.55 | 24.98 | 29.59 | 23.33 | 40.35 | 12.57 | 15.86 | 30.47 | 13.52 | 22.57 | 79.71 |
| *Open-source Remote Sensing Vision-Language Models* | | | | | | | | | | | | |
| VHM | 38.93 | 21.99 | 40.29 | 50.69 | 25.31 | 54.92 | 25.66 | 17.63 | 49.80 | **35.06** | 22.29 | 99.82 |
| SkySenseGPT | 37.76 | 19.06 | 34.98 | 23.33 | 14.02 | 40.48 | 42.47 | 24.95 | 52.58 | 33.10 | 22.50 | 102.8 |
| EarthDial | 42.09 | 23.92 | 42.56 | 67.14 | 46.17 | **123.6** | 29.09 | 25.20 | 85.82 | 21.49 | 15.88 | 90.51 |
| **RSThinker** | **55.69** | **32.29** | **73.55** | **85.12** | **58.88** | 94.81 | **39.82** | **27.17** | **99.83** | 33.96 | 21.19 | **107.5** |

B-4 / MT / Cr: BLEU-4 / METEOR / CIDEr

ing a natural defense against common failure modes. By requiring the model to first ground each object as a distinct entry in its reasoning trace before synthesizing a final tally, the framework inherently mitigates duplication and promotes a more complete search. The consistent, substantial gains across these three related tasks provide strong empirical evidence that the Geo-CoT framework is a key enabler for robust and faithful fine-grained perception.

### 4.2.2 HOLISTIC SCENE UNDERSTANDING: CLASSIFICATION AND CAPTIONING

We then assess the model's ability to interpret the broader context of a scene, addressing whether a methodical, step-by-step reasoning process compromises holistic comprehension. The performance in Scene Classification (Table 6) demonstrates that, on the contrary, the fine-grained analysis fostered by Geo-CoT provides a more robust foundation for high-level understanding. This consistent superiority suggests the model's capacity for systematic evidence gathering translates to a more veridical holistic feature representation. By being trained to ground individual objects and their attributes, the model bases its final classification on a rich, verifiable set of low-level visual facts, rather than relying on potentially spurious correlations in global scene statistics.

This capacity for detailed, fact-based synthesis is further illuminated in Image Captioning (Table 7), where strong performance stems from the Geo-CoT architecture transforming captioning from a monolithic image-to-text mapping into a structured process. The model first grounds key entities and their spatial relationships within its reasoning trace, before synthesizing these grounded elements into a coherent narrative. This mechanism prevents the generation of generic, prototypical captions, instead promoting descriptions rich in detail and verifiably true to the visual evidence. The collective evidence from both tasks indicates that the structured reasoning of Geo-CoT does not hinder, but rather enhances, the model's ability to achieve a profound and accurate understanding of the entire scene.

### 4.2.3 COMPLEX GEOSPATIAL REASONING: VISUAL QUESTION ANSWERING

Finally, we evaluate RSThinker on Visual Question Answering (VQA), where the fine-grained perception and holistic understanding capabilities cultivated previously must converge to resolve complex queries. The architectural advantage of Geo-CoT becomes most salient on queries that necessitate foundational fact-checking. This is demonstrated on the Existence category of VRSBench-VQA (Table 6), where the model's reliability in making a verifiable claim is a direct product of its structured, evidence-grounded reasoning process.

This foundational reliability in evidence verification underpins the model's capacity to execute more complex, multi-step procedures. For comparative queries such as "*Are there more cars near the stadium than near the river?*", the Planning–Grounding–Synthesize framework provides a natural scaffold, compelling the model to first ground each component of the query before synthesizing a final comparative judgment. This consistent performance across the full spectrum of reasoning types—from simple existence checks to complex compositional analysis—reveals that the Geo-CoT framework functions not as a narrow, task-specific solution, but as a general-purpose problem-

Table 8: Ablation study on the impact of CoT-based SFT and GRPO across multiple tasks.

| Models | VG (mIoU) | OC (MAE ↓) | Det (mAP@0.5) | IC (BLEU-4) | SC (Acc) | VQA (Acc) |
|---|---|---|---|---|---|---|
| Base (GLM-4.1V-9B-Base) | 56.26 | 10.81 | 3.56 | 10.99 | 69.78 | 8.16 |
| + SFT (w/o CoT) | 81.80 | 3.272 | 49.36 | 31.14 | 93.33 | 63.57 |
| △ | (+25.54) | (-7.54) | (+45.80) | (+20.15) | (+23.55) | (+55.41) |
| + SFT (w/ CoT) | 87.70 | 2.932 | 74.03 | 33.31 | 96.67 | 74.20 |
| △ | (+31.44) | (-7.88) | (+70.47) | (+22.32) | (+26.89) | (+66.04) |
| + SFT (w/o CoT) + GRPO | 86.47 | 4.510 | 56.77 | 30.87 | 97.56 | 74.09 |
| △ | (+30.21) | (-6.30) | (+53.21) | (+19.88) | (+27.78) | (+65.93) |
| + SFT (w/ CoT) + GRPO | **89.02** | **2.728** | **77.06** | **33.96** | 96.89 | **77.24** |
| △ | (+32.76) | (-8.08) | (+73.50) | (+22.94) | (+27.11) | (+69.08) |

Figure 4: Ablation Study on KL divergence.

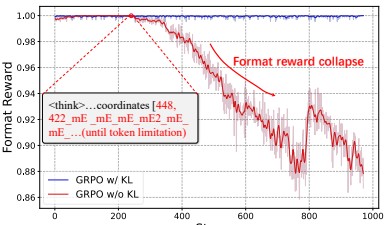

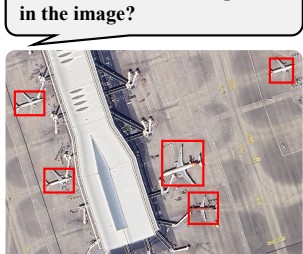

**What is the count of airplanes in the image?**

**RSThinker:** <think>
**(Planning)** To determine the number of airplanes in the image, I start by recognizing that airports typically have multiple aircraft parked on runways or near terminals. The image depicts an airport layout, which provides context for where airplanes are likely to be found. I will systematically examine the various sections of the airport visible in the image. The main terminal area usually has several planes lined up for boarding or unloading passengers. Additionally, I should look at the adjacent runway areas where planes might be waiting for takeoff or taxiing.
**(Grounding)** Upon close inspection of the terminal area at the center of the image, I can identify multiple aircraft parked at the gates. There appear to be several planes aligned parallel to the terminal, each with distinctive shapes and colors, indicating they are indeed airplanes. Next, will count each visible airplane in the identified regions, in total. I can see **three aircraft parked closely together on one side of the terminal**, and **two more on the opposite side**. Lastly, there is **one additional airplane positioned at the far end of the runway**. Therefore, the breakdown of the count is as follows: three airplanes on one side of the terminal, two on the other side, and one on the runway, leading to a cumulative total of six airplanes.
**(Synthesis)** After confirming that all identified targets have the characteristics of airplanes such as wings, fuselage, and tail can conclude that the total number of airplanes present in the image is six. </think>

**RSThinker:** <answer>There are a total of 6 airplanes. </answer>

Figure 5: **Qualitative example of RSThinker's Geo-CoT:** a methodical Planning-Grounding-Synthesis sequence culminating in a justified <answer>.

solving architecture. Ultimately, the VQA results confirm that this architecture seamlessly integrates precise, evidence-based localization with high-level scene interpretation, establishing a new benchmark for robust and complex geospatial reasoning.

### 4.3 ABLATION STUDY

Our comprehensive ablation studies (Table 8) reveal a clear hierarchy of contributions from each framework component. While direct fine-tuning on task-specific data (SFT w/o CoT) yields a significant performance gain over the base model, the introduction of our structured rationales (SFT w/ CoT) unlocks a fundamentally higher performance tier. This substantial performance delta between the two SFT variants stems directly from supervising the model on the computational process itself, rather than merely its final outputs.

The full SFT (w/ CoT) + GRPO model consistently excels, particularly on complex, reasoning-intensive tasks, while applying GRPO without the prerequisite Geo-CoT rationales (SFT w/o CoT + GRPO) proves insufficient to instill the necessary cognitive scaffold. This highlights their symbiotic relationship: rationale-based SFT instills the essential cognitive structure, upon which KL-regularized GRPO subsequently refines the generative policy towards factual correctness. The stabilizing role of KL regularization is visualized in Figure 4, where its absence causes a catastrophic collapse of the learned reasoning format.

### 4.4 QUALITATIVE ANALYSIS

To illustrate the practical implications of our framework, we examine the analytical narrative visualized in Figure 5. The model first constructs a verifiable spatial model by breaking down the total count into its constituent sub-groups (e.g., "*three on one side*", "*two on the opposite*"). This granular evidence, presented within the reasoning trace, directly substantiates the final conclusion. The conclusion is thus rendered verifiable, as it stands as the end-product of a transparent process designed from its inception for methodical analysis. Additional qualitative analysis can be found in Appendix A.6.

**Reasoning from Implicit Intent.** To validate the model's capacity for implicit intent understanding where queries specify functional goals rather than object names, we conducted qualitative experiments on the EarthReason benchmark (Li et al., 2025a). As visualized in Figure 6, in a sociocultural context, a request for a "traditional wedding" venue triggers a systematic search for specific ar-

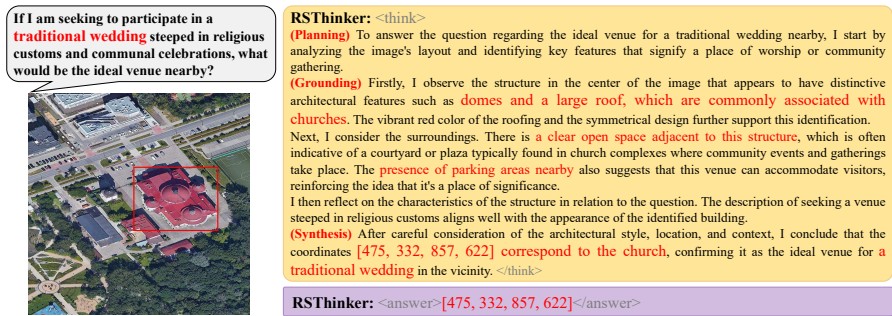

Figure 6: Qualitative results on implicit intent understanding (EarthReason benchmark).

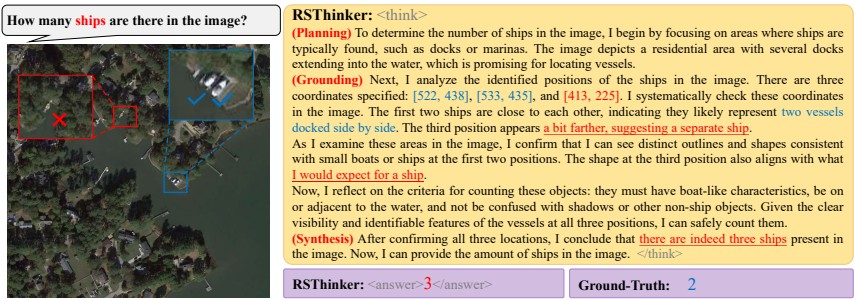

Figure 7: An instance of failure case in object counting. While the reasoning chain is structurally sound and logically coherent, the model misidentifies a non-ship object (red box) as a ship due to visual ambiguity. Crucially, the explicit grounding exposes this error to the user.

chitectural affordances, such as domes and open courtyards, enabling the precise localization of a church. The result demonstrates that RSThinker transcends simple semantic matching, actively reasoning about the functional affordances of geospatial entities to resolve complex, intent-driven queries. Additional examples can be found in Figure 8.

**Failure Analysis.** Despite the robustness of the Geo-CoT framework, Figure 7 reveals a subtle failure mode where the model maintains a coherent reasoning syntax but misidentifies a dock extension as a ship due to textural ambiguity, suggesting that the textual "verification" step can occasionally act as a stylistic heuristic. Crucially, however, the explicit grounding mechanism turns this into a safety feature. Unlike end-to-end baselines that produce opaque errors, RSThinker externalizes the failure by pinpointing the specific bounding box ([413, 225]). This renders the hallucination immediately falsifiable, transforming a potential silent failure into an auditable and interpretable error essential for high-stakes workflows.

## 5 CONCLUSION

In this work, we introduce a framework designed to elicit faithful reasoning in remote sensing Visioned-Language Models. We formalize this reasoning as a Perceptually-Grounded Geospatial Chain-of-Thought (Geo-CoT), where each analytical step must be verifiably grounded in visual evidence. This capability is instilled via a two-stage alignment process, beginning with supervised fine-tuning on Geo-CoT380k, the first large-scale corpus of structured rationales generated for this domain via a novel, scalable pipeline. This SFT-instilled cognitive architecture is then refined via Group Relative Policy Optimization (GRPO), which steers the model's policy toward factually correct final outcomes. While the rationales generated by our pipeline are anchored to ground-truth data, we acknowledge that they may inherit stylistic biases from the generative process itself, a promising avenue for future investigation. Our resulting model, RSThinker, exhibits state-of-the-art outcomes by not only deriving a final answer, but by externalizing the entire verifiable visual interrogation process. Ultimately, this work provides a foundational methodology for developing analytical agents whose reasoning is as verifiable as their final outputs are correct.

## ACKNOWLEDGMENTS

This work was supported by the National Natural Science Foundation of China (Grant No. U22A2098, 62172185), Major Science and Technology Project of Jilin Province (Grant No. 20240212003GX), Major Science and Technology Project of Changchun City (Grant No. 2024WX05), and China Postdoctoral Science Foundation (Grant No. 2025M780312).

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

# A APPENDIX

## A.1 ETHICS STATEMENT

Our research is conducted with a commitment to academic integrity and transparency. All experiments were conducted exclusively on publicly available, open-source remote sensing benchmarks to ensure full reproducibility of our findings. We acknowledge that these datasets may contain inherent geographical or sensor-related biases, a common limitation in the field that could affect the model's generalizability to underrepresented regions. The core objective of our work is to enhance the transparency and verifiability of reasoning processes in VLMs, a goal we believe promotes responsible AI development. We recognize the computational cost associated with training our models and have strived for efficiency. To further promote transparency and enable the community to build upon our work, we commit to the public release of our Geo-CoT380k dataset, model checkpoints, and evaluation code upon publication.

## A.2 REPRODUCIBILITY STATEMENT

We are committed to the full reproducibility of our work and will release all necessary artifacts upon publication. To this end, we will release our primary dataset, Geo-CoT380k, including all structured rationales and data splits, alongside scripts to automatically prepare the cited public benchmarks. This dataset will be accompanied by the full source code for our two-stage alignment strategy, encompassing data preprocessing, SFT, GRPO, and evaluation. To ensure a consistent starting point for replication, the codebase will be bundled with the final model checkpoints for RSThinker and scripts to access the publicly available initialization checkpoint. All of these artifacts will be accompanied by detailed instructions and a requirement file to facilitate the setup of the computational environment and the execution of the full experimental pipeline. After submitting the manuscript, we will simultaneously release the model's weights and related code as open-source.

## A.3 LLM USAGE

This work utilized the large multimodal model GPT-4V (OpenAI, 2023) for the sole and explicit purpose of generating the structured rationales that constitute our Geo-CoT380k dataset. Crucially, the LLM operated within a highly constrained, scalable pipeline designed by the authors, which retrofits verifiable rationales onto ground-truth data to ensure faithfulness by design. The intellectual contribution of this work therefore lies not in the raw output of the LLM, but in the design of the pipeline itself. The formulation of the core methodology, the analysis of experimental results, and the final conclusions presented herein are solely the work of the authors.

## A.4 EXPERIMENTAL SETUP

### A.4.1 TASKS AND DATASETS

To validate the versatility and robustness of RSThinker, we evaluate its performance on a diverse set of canonical remote sensing tasks. These tasks are selected to span the full spectrum from fine-grained perception to holistic scene understanding. To showcase the model's core strengths in systematic, object-level analysis, we first evaluate on object counting using the HRRSD (Zhang et al., 2019), RSOD (Long et al., 2017), DOTAv2-val (Xia et al., 2018; Jian et al., 2019; Ding et al., 2021), and NWPU-VHR (Cheng et al., 2014) datasets, and on object detection across benchmarks such as DOTAv2-val and HRRSD. This precise object-level localization is further tested through visual grounding on the VRSBench-test-VG(Li et al., 2024), DIOR-RSVG (Zhan et al., 2023), RRSIS-D (Liu et al., 2024b) and RSVG (Sun et al., 2022) benchmarks. Moving from object-centric analysis to holistic scene interpretation, we assess performance on scene classification with the NWPU-RESISC45-test (Cheng et al., 2017), AID-test (Xia et al., 2017), WHU-RS19 (Xia et al., 2010), SIRI-WHU (Zhao et al., 2016a;b; Zhu et al., 2016) and UCMerced Yang & Newsam (2010) datasets, and on descriptive image captioning using benchmarks like UCM-Captions (Qu et al., 2016), RSICD (Lu et al., 2017), RSITMD (Yuan et al., 2021), NWPU-captions (Cheng et al., 2022), Sydney-Captions (Lu et al., 2017) and VRSBench-test-cap (Li et al., 2024). Finally, to evaluate the model's ability to handle complex, open-ended queries, we use the challenging VRSBench-test-VQA (Li et al., 2024) and RSVQA-HR-test (Lobry et al., 2020) benchmarks.

Table 9: Comparison of RSThinker with existing generic and RS VLMs on Image Captioning task across multiple benchmarks. B-4, MT, Cr and R-L denote BLUE-4, METEOR, CIDEr and ROUGE-L scores, respectively.

| Method | UCM-Captions | | | | RSICD | | | | RSITMD | | | | NWPU-Captions | | | | Sydney-Captions | | | | VRSBench-cap | | | |
|---|---|---|---|---|---|---|---|---|---|---|---|---|---|---|---|---|---|---|---|---|---|---|---|---|
| | B-4 | MT | Cr | R-L | B-4 | MT | Cr | R-L | B-4 | MT | Cr | R-L | B-4 | MT | Cr | R-L | B-4 | MT | Cr | R-L | B-4 | MT | Cr | R-L |
| *Close-source Commercial Vision-Language Models* | | | | | | | | | | | | | | | | | | | | | | | | |
| Claude-sonnet-4 | 20.12 | 20.99 | 30.04 | 13.35 | 11.58 | 13.90 | 24.57 | 10.63 | 20.14 | 17.15 | 19.31 | 9.13 | 28.32 | 21.98 | 32.46 | 13.38 | 19.85 | 20.14 | 27.55 | 12.52 | 14.62 | 22.36 | 73.49 | 13.81 |
| Gemini-2.0-flash | 9.31 | 6.72 | 13.23 | 5.48 | 10.85 | 8.71 | 21.53 | 9.41 | 15.73 | 9.27 | 17.11 | 7.92 | 20.55 | 11.42 | 22.58 | 9.45 | 31.41 | 24.17 | 38.76 | 16.99 | 14.19 | 22.30 | 86.33 | 13.31 |
| ChatGPT-5 | 28.49 | 25.56 | 40.95 | 17.82 | 16.83 | 16.73 | 34.39 | 15.86 | 27.27 | 21.10 | 29.48 | 14.02 | 39.62 | 25.69 | 48.52 | 20.91 | 28.50 | 24.48 | 39.09 | 17.47 | 18.06 | 25.11 | 88.93 | 15.65 |
| *Open-source Vision-Language Models* | | | | | | | | | | | | | | | | | | | | | | | | |
| MiniGPT-v2 | 25.46 | 19.62 | 30.94 | 13.82 | 15.40 | 12.36 | 26.63 | 12.21 | 25.45 | 16.83 | 25.89 | 11.55 | 37.75 | 19.70 | 35.73 | 15.18 | 26.17 | 17.03 | 23.55 | 12.30 | 26.61 | 18.36 | 68.94 | 16.75 |
| Qwen2.5-VL | 27.87 | 21.48 | 35.36 | 17.23 | 17.80 | 13.72 | 32.19 | 14.62 | 27.92 | 17.24 | 24.90 | 12.20 | 38.89 | 21.40 | 42.11 | 17.75 | 28.60 | 18.77 | 31.81 | 16.87 | 29.21 | 25.01 | 91.84 | 20.29 |
| *Open-source Reasoning Vision-Language Models* | | | | | | | | | | | | | | | | | | | | | | | | |
| Kimi-VL-Thinking | 25.72 | 20.95 | 34.29 | 16.91 | 15.60 | 13.57 | 30.00 | 13.74 | 24.82 | 16.47 | 22.02 | 11.38 | 34.84 | 20.08 | 37.14 | 16.81 | 27.04 | 23.94 | 32.73 | 16.81 | 26.07 | 24.34 | 83.86 | 18.95 |
| GLM-4.1V-Thinker | 20.97 | 22.61 | 33.32 | 15.04 | 12.57 | 15.86 | 30.47 | 13.17 | 20.57 | 19.55 | 24.98 | 11.15 | 29.59 | 23.33 | 40.35 | 16.33 | 20.64 | 22.15 | 29.49 | 13.90 | 13.52 | 22.57 | 79.71 | 13.55 |
| *Open-source Remote Sensing Vision-Language Models* | | | | | | | | | | | | | | | | | | | | | | | | |
| VHM | 42.08 | 27.86 | 66.12 | 25.17 | 25.66 | 17.63 | 49.80 | 20.50 | 38.93 | 21.99 | 40.29 | 18.43 | 50.69 | 25.31 | 54.92 | 22.91 | 44.67 | 35.11 | 67.50 | 23.76 | 35.06 | 22.29 | 99.82 | 24.88 |
| SkySenseGPT | 39.04 | 23.52 | 49.80 | 22.63 | 23.33 | 14.02 | 40.48 | 18.01 | 37.76 | 19.06 | 34.98 | 15.00 | 48.03 | 22.41 | 49.67 | 18.68 | 42.47 | 24.95 | 52.58 | 21.51 | 33.10 | 22.50 | 102.8 | 22.09 |
| EarthDial | 59.77 | 44.08 | 127.7 | 32.43 | 29.09 | 25.20 | 85.82 | 24.19 | 42.09 | 23.92 | 42.56 | 18.35 | 67.14 | 46.17 | 123.6 | 28.96 | 64.04 | 54.91 | 120.9 | 43.75 | 21.49 | 15.88 | 90.51 | 21.40 |
| **RSThinker** | 61.03 | 41.54 | 123.4 | 34.80 | 39.82 | 27.17 | 99.83 | 29.38 | 55.69 | 32.29 | 73.55 | 25.66 | 85.12 | 58.88 | 94.81 | 28.97 | 60.47 | 35.28 | 73.50 | 25.96 | 33.96 | 21.19 | 107.5 | 24.44 |

Table 10: Comparison of RSThinker with existing generic and RS VLMs on Object Detection task across multiple benchmarks.

| Method | DOTA | | | HRRSD | | |
|---|---|---|---|---|---|---|
| | mAP@0.25 | mAP@0.5 | mAP@0.75 | mAP@0.25 | mAP@0.5 | mAP@0.75 |
| *Close-source Commercial Vision-Language Models* | | | | | | |
| Claude-sonnet-4 | 17.80 | 3.89 | 0.29 | 33.82 | 14.87 | 2.80 |
| Gemini-2.0-flash | 35.31 | 14.30 | 4.02 | 62.41 | 28.92 | 4.05 |
| ChatGPT-5 | 20.93 | 8.66 | 1.56 | 50.19 | 13.15 | 0.80 |
| *Open-source Vision-Language Models* | | | | | | |
| MiniGPT-v2 | 8.10 | 3.02 | 1.03 | 16.57 | 9.15 | 4.60 |
| Qwen2.5-VL | 30.57 | 3.18 | 0.50 | 43.48 | 19.04 | 3.34 |
| *Open-source Reasoning Vision-Language Models* | | | | | | |
| GLM-4.1V-Thinker | 47.89 | 40.45 | 30.77 | 70.99 | 55.53 | 36.43 |
| *Open-source Remote Sensing Vision-Language Models* | | | | | | |
| VHM | 7.68 | 2.37 | 0.53 | 17.23 | 12.47 | 7.91 |
| SkySenseGPT | 8.42 | 4.56 | 0.27 | 14.83 | 6.23 | 1.42 |
| EarthDial | 9.42 | 3.52 | 0.00 | 18.13 | 8.05 | 0.59 |
| **RS-Thinker** | **79.56** | **77.06** | **69.88** | **98.54** | **97.99** | **94.59** |

### A.4.2 BASELINES

We benchmark RSThinker against a comprehensive suite of competitive baseline models. These models first include leading proprietary, closed-source systems, such as ChatGPT-5 (OpenAI, 2025), Gemini-2.0-flash (Comanici et al., 2025) and Claude-sonnet-4 (Anthropic, 2025), to establish a performance ceiling against large-scale commercial offerings. Beyond these commercial offerings, our comparison spans open-source models organized along two key axes: their domain specialization (general-purpose versus remote sensing) and their architectural support for explicit reasoning. Our evaluation thus includes leading general-purpose VLMs like MiniGPT-v2 (Zhu et al., 2023) and Qwen2.5-VL (Bai et al., 2025a), alongside their domain-specific remote sensing counterparts such as Geochat (Kuckreja et al., 2024), VHM (Pang et al., 2025), SkysenseGPT (Luo et al., 2024) and EarthDial (Soni et al., 2025). To provide a direct comparison against reasoning-centric approaches, we further include results from both generalist models prompted for CoT and the latest domain-specific reasoning frameworks, namely GLM-4.1V-9B-Thinking (Team et al., 2025b) and Kimi-VL-A3B-Thinking-2506 (Team et al., 2025a).

### A.4.3 IMPLEMENTATION DETAILS AND METRICS

Our implementation of RSThinker is initialized from the GLM-4.1V-9B-Base (Team et al., 2025b) checkpoint and trained on 8 NVIDIA A100 GPUs. During the SFT stage, we train for 3 epochs with a batch size of 32, using the AdamW optimizer with a learning rate of 1e-5. For the subsequent GRPO stage, we finetune for 970 steps, with details of the reward function provided before. Across all experiments, we employ standard, community-accepted metrics to ensure a fair and direct comparison. For object detection and visual grounding, we report mean Average Precision (mAP) and Intersection over Union (IoU). For object counting, we use Mean Absolute Error (MAE). Scene classification and VQA are evaluated on standard Accuracy, while image captioning is assessed using the BLEU-4, Rouge-L, METEOR and CIDEr scores.

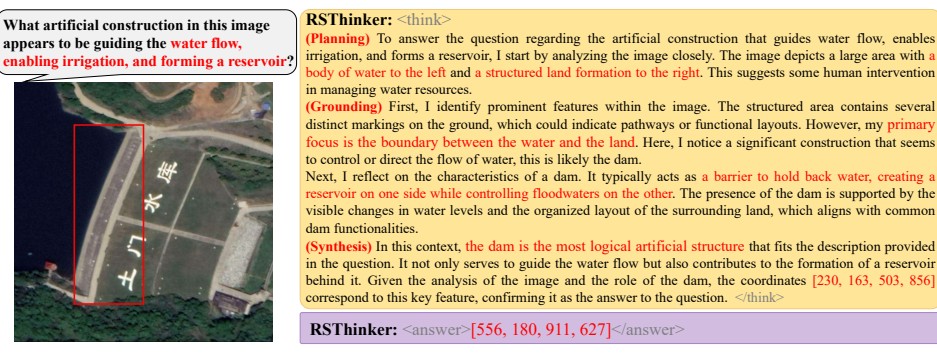

(a) Reasoning for "*restoration of groundwater*" (Pond)

(b) Reasoning for "*Guiding Water Flow*" (Dam)

Figure 8: Qualitative results on implicit intent understanding (EarthReason benchmark).

## A.5 EXPERIMENTAL RESULTS

This section provides the complete experimental tables omitted from the main paper(Tabel 9 and Tabel 10).

## A.6 ADDITIONAL VISUALIZATIONS

This section presents qualitative visualizations of RSThinker's reasoning and predictions across tasks. In Object Detection task(Figure 9), RSThinker first performs a coarse enumeration of aircraft regions and then refines them into precise detection boxes, accurately marking all airplanes in the scene. In Visual Grounding task(Figure 10) with a relational query, "the vehicle on the right of the vehicle on the lower left in the image", RSThinker locates the lower-left vehicle bounding box first and then identifies the target vehicle to its right. For another Visual Grounding task(Figure 11) query, "the building shaped like the number 8", RSThinker detects a structure whose two large loops or curves resemble the key components of the numeral 8, and identifies the correct bounding box. These examples illustrate RSThinker's Planning-Grounding-Synthesize cognitive architecture and its ability to handle relational and shape-centric references. Additional examples are shown in Figure 12 (Image Caption), Figure 13 (Scene Classification), and Figure 14 (VQA).

## A.7 PROMPT FOR CoT GENERATION

We construct Geo-CoT with a two-tier prompting protocol: a shared base prompt that standardizes task intent, input–output format, and our desired Planning-Grounding-Synthesize cognitive architecture, followed by task-specific prompts augmented with a small set of curated in-context exemplars. Auxiliary information (e.g., bounding boxes, referring phrases, spatial attributes, normalized coordinates) is used only during construction to scaffold faithful reasoning and is removed from the released annotations.

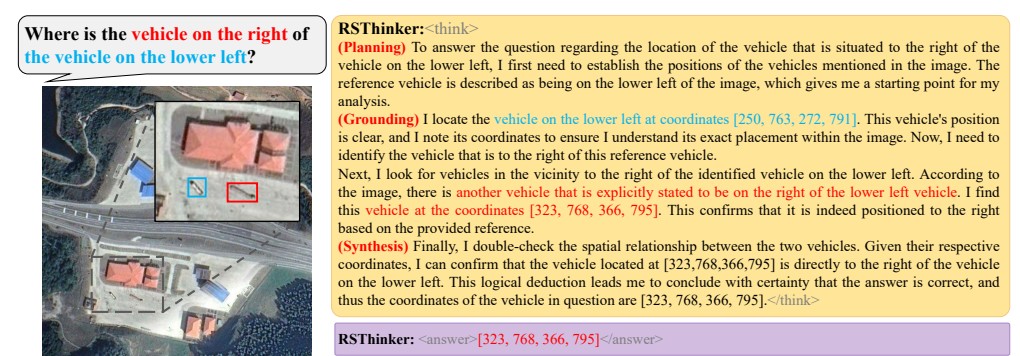

**Detect all airplane in the image.**

**RSThinker:** <think>
**(Planning)** To detect all airplanes in the image, I start by examining the overall layout of the scene, which appears to be an airfield or an airport apron. The image shows a large expanse of concrete and asphalt, typical for aircraft parking and maneuvering. The presence of markings on the ground suggests designated parking areas and taxiways, which are common in airport settings.
The next step is to identify the objects that match the characteristics of airplanes. In satellite imagery, airplanes are generally recognizable by their distinct shapes, including a fuselage and wings. They often contrast with the background due to their size and color, typically appearing lighter against the darker asphalt of the runway or apron.
**(Grounding)** Upon closely analyzing the image, I observe three distinct airplanes. The first airplane is located towards the bottom right quadrant, exhibiting a typical aircraft profile. The second airplane is situated in the middle of the image, slightly angled, which also conforms to expected aircraft geometry. The third airplane is positioned more towards the left side of the image, also identifiable by its shape and orientation. To ensure accurate detection, each airplane's position is noted, and bounding boxes are created to encapsulate their visible extents. The first airplane is captured in the coordinates [661, 522, 854, 645], the second airplane in [415, 549, 594, 673], and the third airplane in [225, 754, 386, 862]. These coordinates reflect the specific locations of the detected airplanes within the image.
**(Synthesis)** In conclusion, I have identified three airplanes at these specified coordinates. I can now provide the detection boxes for all airplanes present in the image. </think>

**RSThinker:** <answer>[[661, 522, 854, 645], [415, 549, 594, 673], [225, 754, 386, 862]]</answer>

Figure 9: **Qualitative Object Detection example of RSThinker's Geo-CoT:** a methodical Planning-Grounding-Synthesis sequence culminating in a justified <answer>.

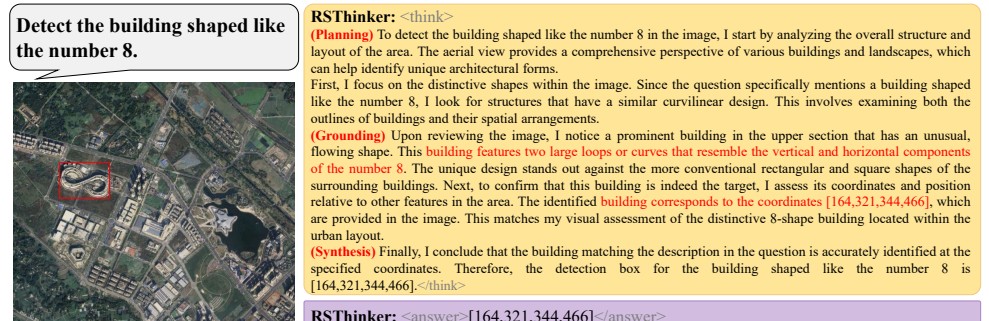

**Where is the vehicle on the right of the vehicle on the lower left?**

**RSThinker:** <think>
**(Planning)** To answer the question regarding the location of the vehicle that is situated to the right of the vehicle on the lower left, I first need to establish the positions of the vehicles mentioned in the image. The reference vehicle is described as being on the lower left of the image, which gives me a starting point for my analysis.
**(Grounding)** I locate the vehicle on the lower left at coordinates [250, 763, 272, 791]. This vehicle's position is clear, and I note its coordinates to ensure I understand its exact placement within the image. Now, I need to identify the vehicle that is to the right of this reference vehicle.
Next, I look for vehicles in the vicinity to the right of the identified vehicle on the lower left. According to the image, there is another vehicle that is explicitly stated to be on the right of the lower left vehicle. I find this vehicle at the coordinates [323, 768, 366, 795]. This confirms that it is indeed positioned to the right based on the provided reference.
**(Synthesis)** Finally, I double-check the spatial relationship between the two vehicles. Given their respective coordinates, I can confirm that the vehicle located at [323,768,366,795] is directly to the right of the vehicle on the lower left. This logical deduction leads me to conclude with certainty that the answer is correct, and thus the coordinates of the vehicle in question are [323, 768, 366, 795].</think>

**RSThinker:** <answer>[323, 768, 366, 795]</answer>

Figure 10: **Qualitative Visual Grounding example of RSThinker's Geo-CoT:** a methodical Planning-Grounding-Synthesis sequence culminating in a justified <answer>.

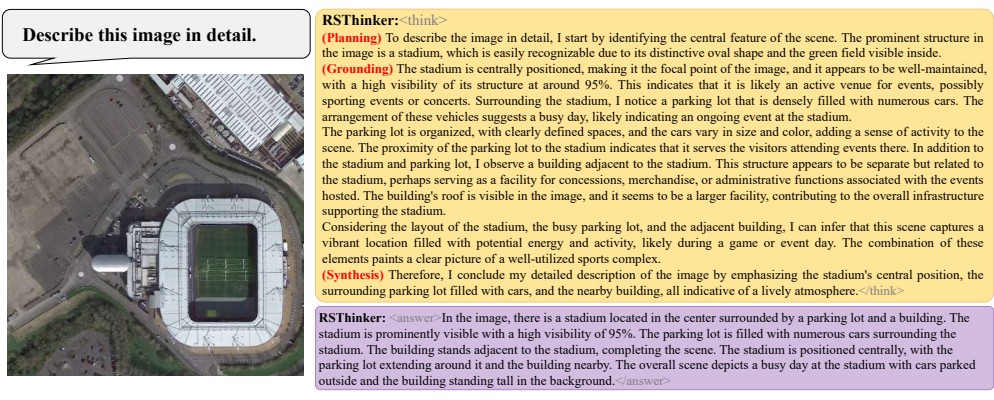

**Detect the building shaped like the number 8.**

**RSThinker:** <think>
**(Planning)** To detect the building shaped like the number 8 in the image, I start by analyzing the overall structure and layout of the area. The aerial view provides a comprehensive perspective of various buildings and landscapes, which can help identify unique architectural forms.
First, I focus on the distinctive shapes within the image. Since the question specifically mentions a building shaped like the number 8, I look for structures that have a similar curvilinear design. This involves examining both the outlines of buildings and their spatial arrangements.
**(Grounding)** Upon reviewing the image, I notice a prominent building in the upper section that has an unusual, flowing shape. This building features two large loops or curves that resemble the vertical and horizontal components of the number 8. The unique design stands out against the more conventional rectangular and square shapes of the surrounding buildings. Next, to confirm that this building is indeed the target, I assess its coordinates and position relative to other features in the area. The identified building corresponds to the coordinates [164,321,344,466], which are provided in the image. This matches my visual assessment of the distinctive 8-shape building located within the urban layout.
**(Synthesis)** Finally, I conclude that the building matching the description in the question is accurately identified at the specified coordinates. Therefore, the detection box for the building shaped like the number 8 is [164,321,344,466].</think>

**RSThinker:** <answer>[164,321,344,466]</answer>

Figure 11: **Qualitative Visual Grounding example of RSThinker's Geo-CoT:** a methodical Planning-Grounding-Synthesis sequence culminating in a justified <answer>.

**Describe this image in detail.**

**RSThinker:** <think>
**(Planning)** To describe the image in detail, I start by identifying the central feature of the scene. The prominent structure in the image is a stadium, which is easily recognizable due to its distinctive oval shape and the green field visible inside.
**(Grounding)** The stadium is centrally positioned, making it the focal point of the image, and it appears to be well-maintained, with a high visibility of its structure at around 95%. This indicates that it is likely an active venue for events, possibly sporting events or concerts. Surrounding the stadium, I notice a parking lot that is densely filled with numerous cars. The arrangement of these vehicles suggests a busy day, likely indicating an ongoing event at the stadium.
The parking lot is organized, with clearly defined spaces, and the cars vary in size and color, adding a sense of activity to the scene. The proximity of the parking lot to the stadium indicates that it serves the visitors attending events there. In addition to the stadium and parking lot, I observe a building adjacent to the stadium. This structure appears to be separate but related to the stadium, perhaps serving as a facility for concessions, merchandise, or administrative functions associated with the events hosted. The building's roof is visible in the image, and it seems to be a larger facility, contributing to the overall infrastructure supporting the stadium.
Considering the layout of the stadium, the busy parking lot, and the adjacent building, I can infer that this scene captures a vibrant location filled with potential energy and activity, likely during a game or event day. The combination of these elements paints a clear picture of a well-utilized sports complex.
**(Synthesis)** Therefore, I conclude my detailed description of the image by emphasizing the stadium's central position, the surrounding parking lot filled with cars, and the nearby building, all indicative of a lively atmosphere.</think>

**RSThinker:** <answer>In the image, there is a stadium located in the center surrounded by a parking lot and a building. The stadium is prominently visible with a high visibility of 95%. The parking lot is filled with numerous cars surrounding the stadium. The building stands adjacent to the stadium, completing the scene. The stadium is positioned centrally, with the parking lot extending around it and the building nearby. The overall scene depicts a busy day at the stadium with cars parked outside and the building standing tall in the background.</answer>

Figure 12: **Qualitative Image Caption example of RSThinker's Geo-CoT:** a methodical Planning-Grounding-Synthesis sequence culminating in a justified <answer>.

| | |
|---|---|
| **Which scene does this image belong to? You must choose one of Parking, Resort … School, Bridge, Church.**
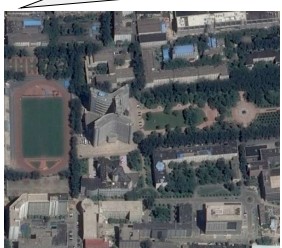 | **RSThinker:** <think>
**(Planning)** First, I examine the overall layout and features present in the image.
**(Grounding)** The image displays a large rectangular building, which is indicative of an educational institution, likely a school. The structure's design includes multiple floors and a spacious outdoor area, suggesting it serves a large number of students.
Next, I notice the presence of a sports field adjacent to the building. This is often a characteristic feature of schools, as they typically have facilities for physical education and sports activities. The field is well-maintained, further indicating it is used regularly by students.
Additionally, there are several trees and landscaped areas surrounding the building, creating a conducive environment for learning. The organized layout of the buildings, along with the recreational spaces, supports the idea that this is an educational setting.
**(Synthesis)** Considering these observations—the large building, sports field, and surrounding green spaces——it is reasonable to conclude that the scene depicted in the image belongs to a school.</think> |
| | **RSThinker:** <answer>School</answer> |

Figure 13: **Qualitative Scene Classification example of RSThinker's Geo-CoT:** a methodical Planning-Grounding-Synthesis sequence culminating in a justified <answer>.

| | |
|---|---|
| **What is the main type of area seen in the image?**
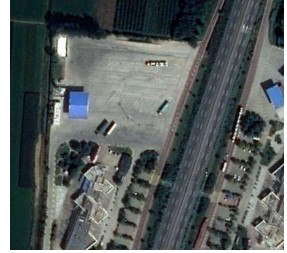 | **RSThinker:** <think>
**(Planning)** To identify the major structure in the image, I first analyze the overall layout captured from an aerial perspective. The image shows a segment of land adjacent to an expressway, which is crucial in determining the nature of the area. The presence of a large, open paved area indicates a facility designed to accommodate vehicles, suggesting a service area for travelers on the expressway.
**(Grounding)** Upon closer inspection, I notice several parked vehicles scattered throughout this area, which reinforces the idea that this structure is indeed a service area meant for rest or refueling. The vehicles vary in size and position, indicating active use of the facility. Notably, there is a larger vehicle located towards the center of this area, further supporting the notion of functionality associated with the service area.
Additionally, the blue-roofed structure visible in the image is characteristic of service areas, often providing amenities for drivers. The surrounding green vegetation and the expressway running parallel to this area provide context, confirming that this is a service area designed for convenience along a major route.
**(Synthesis)** Considering all these elements, it is clear that the most prominent structure depicted in the image is the expressway service area, characterized by its layout, the presence of vehicles, and supportive infrastructure. Therefore, I can conclude that the major structure in the image is indeed the expressway service area.</think> |
| | **RSThinker:** <answer>expressway-service-area</answer> |

Figure 14: **Qualitative VQA example of RSThinker's Geo-CoT:** a methodical Planning-Grounding-Synthesis sequence culminating in a justified <answer>.

---

**Shared base prompt (prepended to all tasks).**

You are an expert in the field of remote sensing with strong reasoning abilities, capable of identifying, analyzing, and inferring information in remote sensing images. Now you need to help construct a CoT dataset in the field of remote sensing, to help other models think better. For each remote sensing image, utilize the provided auxiliary information to better understand the image. Now construct a detailed remote sensing TASK-CoT dataset.

For each sample, you will receive ** image **, ** auxiliary information **, ** question **, and** correct answer **. You need to carefully understand the ** image ** to facilitate the capture of key targets in the image by ** auxiliary information ** (Note: auxiliary information is correct, but may not be comprehensive, you need to combine it with your own understanding to identify all the information of the image). For the problem, fully understand the correct answer, and think of the solution path to the answer, integrating it into the general CoT from question to answer.

Importantly, auxiliary information is to help you understand the image and not let it appear in CoT. Even if you do not agree with certain viewpoints of the auxiliary information, you can ignore these viewpoints, but do not mention them in the CoT.

Note that the CoT should infer the answer from the problem, and the correct answer should not appear abruptly in advance. Therefore, statements such as 'this is consistent with the correct answer' should not be included in CoT. The CoT should include a process of thinking and deduction, as well as a process of reflection. The CoT should naturally be divided into several segments, with each segment separated by a "\n\n\n".

Next, provide few examples of TASK-CoT.

{Task-specific exemplars}

Now, please construct a TASK-CoT for the input. Please carefully examine the image, read and understand the question and correct answer, think carefully, deduce the thought process from question to answer, and organize it into a CoT presentation. The CoT format is "CoT": "CoT content" . Do not return any additional information.

The base prompt instantiates a Planning–Grounding–Synthesize cognitive architecture: first decompose the task into tractable subgoals, then ground each step in observable, object/region-level evidence, and finally synthesize a concise answer after explicit verification. It forbids unverifiable claims and requires explicit reference to evidence when applicable (e.g., bounding boxes, coordinates, directions, relative size/position). We implement the annotator with GPT-4V (OpenAI, 2023) under constrained prompt, and employ in-context learning with a few high-quality exemplars to reinforce Planning–Grounding–Synthesize style reasoning. Minor task-specific variants of the base prompt are used to explicitly cue the current task while keeping the core instructions unchanged.

For each task, we append a minimal task-specific template to the shared base prompt and supply few curated in-context exemplars. These exemplars are chosen to span diverse scene types and difficulty levels (including edge cases) and must strictly follow the canonical output format to ensure reliable parsing and consistent reasoning.

---

**Object counting task exemplar**

Next, provide two examples of count-CoT.
The first example for a smaller number of targets, maybe less than 8-10.
Input:

```
{
"question": "What is the amount of ship in the image? \textbackslash
    nAnswer the question using a single word or phrase.",
"auxiliary information":{
    "image_size": [800,800],
    "objects": {
        "ship_position": [[612, 761], [628, 705], [657, 531]],
        "harbor_position": [[492, 715], [527, 504], [568, 8]]
    },
    "count": {
        "ship": 3,
        "harbor": 3
    }
},
"answer": "3",
}
```

Output: { "CoT": "To determine the number of ships in the image, we begin by identifying the most likely locations for ships—typically near docks, piers, or water edges. This image shows a coastal residential area with multiple piers extending from private properties into the water.\n\nWe systematically analyze the image from top to bottom. The topmost dock has no visible ship. The second dock from the top clearly has a small white vessel at its end, identifiable by its sharp shape, white color, and shadow consistent with a boat. The third dock—closer to the bottom of the image—shows two vessels: a large white yacht with a visible bow and deck features, and a second boat partially covered with a blue tarp next to it.\n\nSo all the ship targets we found in the image are as follows: a small white vessel at the end of second dock, two vessols at the third dock—a large white yacht with a visible bow and deck features and a second boat partially covered with a blue tarp next to it.\n\n Now we cross-verify the water area away from the docks. No additional vessels are observed floating independently or anchored elsewhere. All identifiable ships are docked at the piers, suggesting no free-floating vessels.\n\nTo ensure accuracy, we reflect on the identification criteria of 'ship': the objects must have boat-like geometry, orientation on water, and features such as hulls, decks, or covers. All three identified objects meet this criterion. We do not count objects such as floating platforms or shadows as ships. Now, I can provide the amount of ships in the image." }
The first example end.
The Second example is a larger number of targets, maybe more than 8-10. For the large number of targets, The count-CoT should count by region, identify all regions in the image where targets exist, and provide **the exact number of targets in each region**, and finally count the number of targets.
Input:

```
{
  "question": "how many small-vehicles are there in the image?\
   nAnswer the question using a single word or phrase.",
  "auxiliary information": {
    "image_size": [800, 800],
    "objects": {
      "small-vehicle_position": [
        [695,365],[723,335],[752,307],[781,278],[796,162],
        [745,216],[770,188],[715,244],[329,791],[326,754],
        [664,395],[619,443],[584,479],[549,515],[519,546],
        [490,574],[462,605],[436,630],[408,660],[381,687],
        [352,688],[687,274],[660,302],[623,338],[596,364],
        [571,393],[540,423],[512,452],[484,482],[455,511],
        [427,539],[392,577],[361,607],[334,633],[263,695],
        [201,679],[168,666],[134,657],[89,652],[49,650],
        [12,656],[641,421]
      ],
      "swimming-pool_position": [[463,687]]
    },
    "count": { "small-vehicle": 42, "swimming-pool": 1 }
  },
  "answer": "42"
}
```

Output: { "CoT": "To answer the question regarding the number of small vehicles in the image, I begin by carefully inspecting the urban area depicted in the satellite image. The term 'small-vehicle' generally refers to passenger cars or similar-sized transportation units, which typically appear on roads, in parking areas, or near residential and commercial buildings.\n\nScanning the image, I identify several regions where such vehicles are likely to be found. The straight road along the right edge of the image contains two dense row of parked vehicles aligned parallel to the curb. These vehicles are of consistent size and shape, suggesting they are indeed small vehicles. Carefully count their number as 17 vehicles above the road and 16 vehicles below the road. Additionally, more vehicles are visible around the curved road at the bottom-left portion of the image. I can count 9 vehicles at the cured road. I must also check for any vehicles near building entrances or driveways, as small vehicles might be parked or partially obscured there.\n\nSo all the ship targets we found in the image are as follows: 17 vehicles above the right road, 16 vehicles below the right road, 9 vehicles at the cured road.\n\nTo ensure completeness, I systematically count each visible vehicle in these areas. I mark each location where a vehicle is visible—most are clearly distinguishable by their rectangular shape, shadow pattern, and contrast with the road surface. Even in areas where vehicles are clustered or partially covered by trees or nearby structures, their general size and alignment help confirm their identity as small vehicles.\n\nAfter completing a detailed inspection and verification of all visible small vehicles in the image, I conclude the total number of small vehicles present. Now, I can provide the amount of small vehicles in the image." }

The second example end.

## Image caption task exemplar

Next, provide an example of caption-CoT.
Input:

```
{
"question": "Describe the image in detail.",
"auxiliary information": {
      "objects": [
      {
          "obj_id": 0,
          "referring_sentence": "The toll station is positioned at
   the center of the image.",
          "obj_cls": "expressway-toll-station",
```

```
                "obj_coord": [
                    0.45,
                    0.43,
                    0.59,
                    0.59
                ],
                "obj_position": "center",
                "obj_rel_position": "",
                "obj_size": "small",
                "obj_rel_size": "",
            },
            {
                "obj_id": 3,
                "referring_sentence": "The right-most small vehicle is
        located near the center, to the right of the toll station.",
                "obj_cls": "vehicle",
                "obj_coord": [
                    0.58,
                    0.44,
                    0.62,
                    0.47
                ],
                "obj_position": "center",
                "obj_rel_position": "right-most",
                "obj_size": "small",
                "obj_rel_size": "",
            }
        ]
    },
    "answer": "The image, sourced from GoogleEarth, shows a rural area
        with an expressway-toll-station situated at the center.
        Alongside the toll station, multiple small vehicles are visible,
         with one positioned on the middle-left and another that is the
        right-most in the center relative to the toll station.",
}
```

Output: { "CoT": "To describe the image in detail, I begin by identifying key man-made structures and their surroundings. The image appears to be taken from a satellite or aerial perspective, capturing a segment of a road that intersects the center of the image. A distinctive blue-roofed structure is positioned along the road, which is indicative of a toll station, given its placement and function at a narrow point in the road. This structure is centrally located and stands out due to the contrast of its blue roof against the surrounding green terrain. \n\n\nNext, I observe the presence of several small vehicles on the road. These vehicles are distinguishable by their size and light-colored shapes. At least two vehicles are clearly visible: one is to the left of the toll station (middle-left in the image), and another is to the right of the toll station, as the right-most vehicle near the center. Their relative positions to the toll station confirm their function as vehicles in transit or waiting. \n\n\nAdditionally, below the toll station, there is a rectangular area containing a building with a red roof and surrounding features like shadows and parked cars, suggesting it is a nearby facility related to the toll station or local administration. The walls around this structure further support that this is a standalone, purpose-built building, likely a residential or administrative facility. \n\n\nConsidering the layout, objects, and spatial relationships, I deduce that this is a rural location due to the surrounding undeveloped green landscape and limited infrastructure. The toll station and vehicles are the primary indicators of human activity. After reflecting on the structures visible in the image, I conclude a comprehensive and detailed description must include these components—the rural environment, central toll station, nearby vehicles, and the presence of a building below the toll station. Now, I can provide a detailed description about this image. " }

The example end.

---

**Object Detection task exemplar**

Next, provide two examples of Object Detect-CoT.

The first example for a smaller number of targets, maybe less than 8-10.

Input:

```
{
    "question": "Detect all airplane in the image.",
    "answer":
[[703,252,805,345],[835,487,985,582],[855,734,936,793]],
    "auxiliary information":{
        "objects": {
                "airplane":
[[703,252,805,345],[835,487,985,582],[855,734,936,793]],
                "ground track field": [[38,11,383,394]]
            },
        "count": {
            "airplane": 3,
            "ground track field": 1
        }
    },
}
```

Output: { "CoT": "To detect all airplanes in the image, we first need to understand the overall layout and characteristics of the scene. From a top-down perspective, the image appears to depict a large airport or airbase facility. On the right side of the image, there is a wide expanse of concrete consistent with an airport apron or taxiway, commonly used for aircraft parking, loading, and movement. The left portion of the image includes multiple buildings, greenery, and a ground track field, suggesting mixed-use infrastructure possibly associated with aviation operations or nearby institutional facilities.\n\n\nThe task is to locate all airplanes within the image. From a visual perspective, airplanes in satellite imagery are generally characterized by a fuselage, wings extending laterally, and sometimes visible tail sections or shadows. They are usually white or light gray and stand out against the concrete apron due to their shape and symmetry. \n\n\nCarefully scanning the apron area on the right side of the image, three distinct airplanes can be observed. The first airplane is near the top right quadrant of the apron, oriented roughly northwest to southeast. The second airplane is positioned slightly below center, larger in size, and is placed near the midsection of the apron. The third airplane is in the bottom right quadrant, near the edge of the apron area. These three objects clearly exhibit the typical geometry of airplanes when viewed from above and are isolated from other vehicles or structures, confirming their identification.\n\n\nAfter identifying and verifying the airplane locations, we convert their positions into coordinate ranges in the thousandths scale relative to the image dimensions. The bounding boxes for the airplanes are as follows: the first airplane is near the top right quadrant of the apron at [703,252,805,345], the second airplane is positioned slightly below center at [835,487,985,582], and the third airplane is in the bottom right quadrant at [855,734,936,793]. These coordinates accurately encompass the visual extents of each detected airplane. Thus, the image contains exactly three airplanes at these specified locations. Now I can provide the detection box for all airplanes in the image."
}

The first example end.

The Second example is a larger number of targets, maybe more than 8-10. For the large number of targets, The Object Detect-CoT should count by region, identify all regions in the image where targets exist, and provide **the exact targets in each region**, and finally provide the detection box for all targets.

Input:

```
{
    "question": "Detect all small-vehicle in the image.",
    "auxiliary information":{
        "objects": {
                "small-vehicle": [
                    [182,325,205,338], [263,268,288,278],
                    [212,282,236,295], [100,806,123,828],
```

```
                      [791,122,812,141], [775,93,796,115],
                      [530,301,552,318], [341,81,358,102],
                      [592,295,620,315], [478,342,497,355],
                      [498,332,522,348], [52,296,78,313],
                      [132,258,161,271], [738,121,756,138],
                      [782,71,808,92]
                  ],
                  "storage-tank": [[317,5,387,65]]
              },
          "count": {
              "small-vehicle": 15,
              "storage-tank": 1
          }
      } ,
      "answer": [
              [182,325,205,338], [263,268,288,278], [212,282,236,295],
              [100,806,123,828], [791,122,812,141], [775,93,796,115],
              [530,301,552,318], [341,81,358,102], [592,295,620,315],
              [478,342,497,355], [498,332,522,348], [52,296,78,313],
              [132,258,161,271], [738,121,756,138], [782,71,808,92]
          ],
}
```

Output: { "CoT": "To detect all small vehicles in the image, I begin by scanning the image systematically from top-left to bottom-right, focusing on regions that contain roads, intersections, or parking areas, as these are the most likely locations for small vehicles. In remote sensing imagery, small vehicles are typically identifiable by their small rectangular or oval shapes, high contrast with road surfaces, and regular spacing when parked.\n\n\nThe first noticeable region is near the top of the image, where a broad road runs horizontally across. Within this region, multiple small objects are visible that match the expected characteristics of small vehicles. These objects have rectangular shapes and are aligned along traffic lanes or parking areas. Carefully examining this section reveals multiple vehicles, including one near the center and two others slightly left and right. A curved structure is also visible, but it is not a vehicle, so I ignore it.\n\n\nNext, I look at the middle and lower-middle sections of the image. A prominent bright structure (possibly a roof or overpass) obscures part of the area, but beneath and around it, vehicles are visible. There are several parked or moving vehicles, particularly clustered near entrances and exits of adjacent roads. At least four vehicles are evident by their size, color contrast, and position relative to the road.\n\n\nThe lower-left section of the image contains another road with curved lanes and an exit area. Vehicles can be seen parked or moving along this segment as well. These vehicles are again identified by their compact shapes and positions aligned with road markings. A vehicle appears at the bottom left corner, and another is slightly above it.\n\n\nFinally, I inspect the far-right region and the top-right quadrant. This area includes a continuation of the road network, and I identify several more vehicles, particularly at intersections and nearby parking zones. These are less occluded and clearly match the known signature of small vehicles.\n\n\nIn total, I identify 15 small vehicles distributed across various regions of the image: Top central road region: 3 vehicles; Mid-upper right road network: 4 vehicles; Under the curved roof structure: 3 vehicles; Bottom-left road area: 2 vehicles; Bottom-central area: 1 vehicle; Upper-left road: 2 vehicles. Each of these vehicles is matched to a detection box based on their visible boundaries. After bounding each vehicle, I obtain the following bounding boxes:[[182, 325, 205, 338], [263, 268, 288, 278], [212, 282, 236, 295], [100, 806, 123, 828], [791, 122, 812, 141], [775, 93, 796, 115], [530, 301, 552, 318], [341, 81, 358, 102], [592, 295, 620, 315], [478, 342, 497, 355], [498, 332, 522, 348], [52, 296, 78, 313], [132, 258, 161, 271], [738, 121, 756, 138], [782, 71, 808, 92]]. Now I can provide the detection box for all small vehicles in the image." }

The second example end.

---

**VQA task exemplar**

Next, provide an example of VQA-CoT.
Input:

```
{
    "question": "Are the vehicles predominantly parked on the left
    or right side of the image?",
    "answer": "left",
    "type": "object position",
    "auxiliary information":{
        "caption": "This high-resolution image from GoogleEarth
    depicts a parking area with several vehicles and distinguished
    by a combination of asphalt surfaces and surrounding greenery.
    In total, there are nine small vehicles, distributed in
    different positions throughout the image. Notable objects
    include trees casting shadows and a building located on the left
     side, indicating a semi-urban environment. All vehicles are
    parked, with many situated along the edges of the lot.",
        "objects": [
            {
                "obj_id": 4,
                "referring_sentence": "The vehicle located at the
    top-middle of the image.",
                "obj_cls": "vehicle",
                "obj_coord": [
                    0.52,
                    0,
                    0.55,
                    0.06
                ],
                "obj_position": "top-middle",
                "obj_rel_position": "top-most",
                "obj_size": "",
                "obj_rel_size": ""
            }
        ]
    }
},
```

Output: "CoT": "To determine whether the vehicles are predominantly parked on the left or right side of the image, we must first clearly define what constitutes the 'left' and 'right' regions. In this context, the image is oriented with the top representing the north, meaning the left and right refer to the western and eastern parts of the image, respectively.\n\nNext, I begin by visually scanning both sides of the image. I identify and count the number of parked vehicles on each side. On the left side of the image (the western half), I can observe several vehicles parked near the building and along the road. Specifically, there are 9 vehicles clearly visible parked in this region, of which 3 vehicles are next to buildings and 6 vehicles are on the road. On the right side (the eastern half), I observe significantly fewer vehicles, with only around 2 vehicles clearly parked, and possibly one more partially visible near the lower edge.\n\nHaving made these observations, I can now compare the counts. The left side shows a denser concentration of parked vehicles compared to the right. Moreover, the spatial distribution of these vehicles confirms this trend — the majority of parking appears clustered in the upper-left quadrant of the image, near the building and shaded area. The right side, in contrast, is more open and has fewer parked cars.\n\nTherefore, through a visual examination and vehicle count comparison across both sides of the image, I conclude that the vehicles are predominantly parked on the left side."
The example end.

---

---

**Scene classification task exemplar**

Next, provide an example of Scene Classification-CoT.
Input:

```
{
    "question": "Classify the given image in one of the following
    classes. Classes: parking, baseballfield, school, resort,
    industrial, square, commercial, playground, port, mountain,
    forest, bareland, farmland, desert, denseresidential, airport,
    meadow, mediumresidential, center, storagetanks, park, bridge,
    church, pond, railwaystation, viaduct, stadium, beach, river,
    sparseresidential. \nAnswer in one word or a short phrase.",
    "answer": "airport",
}
```

Output: { "CoT": "First, I observe the overall layout and features present in the image. The image shows a large built structure with a highly organized layout and multiple identical objects aligned along both sides of a central building. The structures extending from the central building resemble jet bridges or concourses, which are commonly found in airports for boarding and deplaning passengers from aircraft.\n\nNext, I examine the objects that are aligned with these structures. These appear to be commercial aircraft, identifiable by their wings, fuselage, and tail configurations. The high number of aircraft, their close positioning to the terminal building, and the presence of taxiways and runways strongly suggest that this location is used for air travel operations.\n\nI also notice a network of runways, taxiways, and apron areas typical of an airport's design. The large paved areas around the terminal allow for aircraft movement and parking. The presence of service roads, ground vehicles, and gate markings further reinforces this being an operational airport.\n\nGiven the presence of a terminal structure, multiple commercial aircraft, and surrounding airport infrastructure such as taxiways and aprons, the most fitting classification among the provided categories is airport." }
The example end.

---

**Visual grounding task exemplar**

Next, provide an example of VG-CoT.
Input:

```
{
    "question": "where is the vehicle situated towards the bottom-
    right side of the image is on the road parallel to the overpass
    .",
    "answer": [790,820,820,860],
    "auxiliary information":{
        "caption": "This high-resolution image from GoogleEarth
    depicts a parking area with several vehicles and distinguished
    by a combination of asphalt surfaces and surrounding greenery.
    In total, there are nine small vehicles, distributed in
    different positions throughout the image. Notable objects
    include trees casting shadows and a building located on the left
     side, indicating a semi-urban environment. All vehicles are
    parked, with many situated along the edges of the lot.",
        "objects": [
            {
                "obj_id": 0,
                "referring_sentence": "The overpass runs diagonally
    across the image and casts a shadow to its right.",
                "obj_cls": "overpass",
                "obj_coord": [
                    390,
                    530,
                    1060,
```

```
                    960
                ],
                "obj_position": "",
                "obj_rel_position": "",
                "obj_size": "small",
                "obj_rel_size": ""
            },
            {
                "obj_id": 1,
                "referring_sentence": "The vehicle situated towards
    the bottom-right side of the image is on the road parallel to
    the overpass.",
                "obj_cls": "vehicle",
                "obj_coord": [
                    790,
                    820,
                    820,
                    860
                ],
                "obj_position": "bottom-right",
                "obj_rel_position": "",
                "obj_size": "small",
                "obj_rel_size": ""
            },
            {
                "obj_id": 2,
                "referring_sentence": "The vehicle located on the
    road below the overpass is near the bottom-middle of the image
    .",
                "obj_cls": "vehicle",
                "obj_coord": [
                    480,
                    790,
                    530,
                    830
                ],
                "obj_position": "bottom-middle",
                "obj_rel_position": "left-most",
                "obj_size": "small",
                "obj_rel_size": ""
            },
            {
                "obj_id": 3,
                "referring_sentence": "The vehicle on the road above
     the overpass is positioned towards the top-right of the image
    .",
                "obj_cls": "vehicle",
                "obj_coord": [
                    780,
                    280,
                    800,
                    320
                ],
                "obj_position": "top-right",
                "obj_rel_position": "top-most",
                "obj_size": "small",
                "obj_rel_size": ""
            }
        ]
    }
},
```

Output: { "CoT": "To address the question, I must first understand the spatial layout of the image. The question refers to a vehicle towards the bottom-right side of the image that is situated on a road running parallel to an overpass. This indicates a relationship between the location of the vehicle and the orientation of the overpass.\n\n\nUpon examining the image, I observe a large overpass running diagonally from the lower-left to the upper-right. The overpass crosses above two visible roadways, one above and one below it, each carrying vehicular traffic. The shadow of the overpass is cast to the right side, confirming the orientation.\n\n\nNext, I identify multiple vehicles in the image. The target vehicle must be towards the bottom-right of the image and located on a road that runs roughly in parallel to the overpass. Looking in the bottom-right quadrant of the image, I spot a small vehicle traveling along a lane that follows the same general diagonal direction as the overpass. This suggests the road is parallel to the overpass and supports the condition stated in the question.\n\n\nFinally, I verify that this vehicle's position aligns with the described location—bottom-right—and confirm it is situated on the correct road. This process ensures that my identification is accurate. Thus, the coordinates [790,820,820,860] correctly correspond to the vehicle referenced in the question." }
The example end.

