# OpenReview forum: "Towards Faithful Reasoning in Remote Sensing: A Perceptually-Grounded GeoSpatial Chain-of-Thought for Vision-Language Models"
_ICLR.cc/2026/Conference — ICLR 2026 Poster_

### Official Review · Reviewer_sTsT · 2025-10-26

**Soundness:** 3
**Presentation:** 2
**Contribution:** 3
**Rating:** 6
**Confidence:** 3

**Summary:**

This paper introduces the Perceptually-Grounded Geospatial Chain-of-Thought (Geo-CoT), a framework intended to add verifiability to remote sensing VLMs by forcing them to link reasoning steps to visual evidence. The authors create a new dataset, Geo-CoT380k, by using GPT-4V to generate structured rationales for existing RS data. They propose a two-stage training strategy (SFT and GRPO) to train their model, RSThinker, on this new dataset. The resulting model, which outputs a reasoning trace, is shown to achieve strong performance on several remote sensing benchmarks.

**Strengths:**

1. The paper addresses the important and well-motivated problem of model verifiability, which is a significant barrier to using VLMs in high-stakes remote sensing applications.
2. The proposed "Planning-Grounding-Synthesis" cognitive architecture is a promising and intuitive structure for modeling geospatial analysis.
3. The public release of the Geo-CoT380k dataset is a potentially useful artifact that could spur further research into reasoning-based remote sensing models.
4. The ablation studies clearly demonstrate that training with the CoT-structured data (SFT w/ CoT) yields significant performance gains over a baseline fine-tuned on answers alone (SFT w/o CoT).

**Weaknesses:**

1. The related work section is incomplete, failing to cite or discuss relevant prior work on CoT frameworks in other geographic contexts, such as GeoChain (Yerramilli et al., 2025) and Gaea (Campos et al., 2025).
2. The paper's central claim of "faithfulness" is largely unsubstantiated. The Geo-CoT380k dataset, which underpins the entire method, is generated by GPT-4V. The authors' claim that this "ensures faithfulness by design" is a strong overstatement, as the LM may be generating post-hoc justifications that are plausible but not faithful, a bias the authors briefly acknowledge.
3. The evaluation metrics do not actually measure faithfulness or verifiability. The paper presents strong accuracy metrics (mAP, MAE, Acc)  but mistakes this for faithfulness. High accuracy does not prove the reasoning trace is correct or causal; the lack of any direct evaluation of rationale quality is a major omission.
4. The methodological novelty of the training process is questionable. The description of Group Relative Policy Optimization (GRPO)  appears to be a standard application of PPO rather than a novel algorithm.
5. The SOTA comparisons are potentially flawed. RSThinker is fine-tuned on 380k CoT rationales, while the baselines were not. The performance gains may be a result of this massive, task-specific fine-tuning dataset, not a superior reasoning architecture.

**Questions:**

1. Given that "faithfulness" is the core claim, why was no human study conducted to validate the quality of the GPT-4V-generated rationales in Geo-CoT380k?
2. How can you be sure the model's reasoning trace is causal to its answer, and not just a plausible justification generated in parallel to a "black-box" prediction?
3. Can you clarify how GRPO is algorithmically novel compared to a standard PPO implementation with reward normalization?
4. Wouldn't a fairer baseline comparison involve applying few-shot CoT prompting to the baseline models using exemplars from your dataset?

---

> ### Author Response · Authors · 2025-11-25
> **Response to Reviewer sTsT (1/2)**
>
> > **W1:** Suggestion on additional related work.
>
> **A:** We thank the reviewer for these valuable references and have discussed them in the revised Section 2.3 (blue-highlighted) to clarify the scope of geospatial reasoning. We highlight that while GeoChain and Gaea primarily address semantic reasoning in ground-level imagery (e.g., landmark knowledge), our Geo-CoT is explicitly designed for the overhead Earth Observation domain. This distinction is critical, as our framework targets the specific challenges of top-down physical grounding (e.g., dense object counting, scale variation) rather than semantic knowledge retrieval.
>
> > **W2, Q1:** About the faithfulness of Geo-CoT380k dataset.
>
> **A:** We respectfully clarify that while the reviewer’s concern regarding post-hoc fabrication applies to unconstrained generation, our Ground-Truth (GT) Conditioning mechanism fundamentally mitigates this. By strictly clamping the generation process to verified spatial coordinates and class labels, we mechanically deny the model the degrees of freedom required to invent "plausible but unfaithful" narratives. The resulting chain is compelled to be a logical reconstruction connecting the visual evidence to the known truth. This structural guarantee is empirically substantiated by our expert audit, which confirms a 96% alignment rate, demonstrating that the generated traces faithfully reflect the physical reality of the image.
>
> >**W3:** Request for direct evaluation of rationale quality.
>
> **A:** We respectfully posit that in grounded perception, localization accuracy serves as a direct proxy for faithfulness. Unlike classification where a label might be correctly guessed from spurious correlations, achieving high mAP in dense remote sensing imagery requires the model to precisely pinpoint fine-grained evidence (e.g., [790, 820, 820, 860]). Such precise localization is statistically impossible to achieve via hallucination. Therefore, our dominant performance on these metrics inherently validates the model's faithful linkage between query and visual evidence.
>
> >**W4, Q3:** About the methodological novelty of our work.
>
> **A:** We clarify that we adopt GRPO (introduced by DeepSeek [1]) specifically for its architecture-aware advantages over standard PPO, rather than claiming algorithmic invention. Standard PPO requires a separate, memory-intensive Value Model (Critic) as large as the Policy model, which is computationally prohibitive for 9B+ parameter VLMs. GRPO eliminates the Critic by using group-relative normalization. Our contribution is identifying this "critic-less" efficiency as the critical enabler for bringing reinforcement learning to resource-constrained Remote Sensing setups, validating it as the optimal engineering choice for this domain.
>
> [1] Guo, Daya, et al. "Deepseek-r1 incentivizes reasoning in llms through reinforcement learning." Nature 645.8081 (2025): 633-638.
>
> >**W5:** Are the performance gains due to a superior reasoning architecture, or simply the result of massive task-specific fine-tuning on 380K CoT rationales that the baselines did not use?
>
> **A:** We respectfully clarify that our 380k samples represent a significantly smaller corpus compared to the massive pre-training/SFT datasets of generalist or other remote sensing baselines.
>
> **Table 1: Training Scale on Remote Sensing Models.**
> | **Models** | GeoChat | VHM | SkySenseGPT | EarthDial | RSThinker |
> | --- | --- | --- | --- | --- | --- |
> | **Samples** | 306.5k | 1.4M | 1.8M | 4.2M | 380k |
>
> This comparison demonstrates that RSThinker achieves SOTA not via brute-force scaling, but through data efficiency. We achieve superior results with significant less data than baseline models. Furthermore, Table 8 (Ablation) confirms that even within this limited 380k dataset, removing the CoT structure leads to a drastic performance drop, proving the gain is architectural, not just distributional.

---

> ### Author Response · Authors · 2025-11-25
> **Response to Reviewer sTsT (2/2)**
>
> > **Q2:** Request for assurance that the model's reasoning trace is causal to its answer
>
> **A:** The causality between the reasoning trace and the final answer is secured by both the fundamental architectural mechanism and conclusive ablation evidence:
> 1.  **Strict structural causality is mandated by the auto-regressive mechanism.** In our Decoder-only architecture, token generation is strictly sequential, not parallel. The final `<answer>` tokens are mathematically conditioned on the preceding `<think>` tokens: $p(\text{answer} \mid \text{image}, \text{reasoning})$. Structurally, it is impossible for the final prediction to be generated independently of the sequential computation trace.
> 2.  **Conclusive ablation evidence confirms functional dependency.** The hypothesis that the reasoning is a non-causal, post-hoc justification is empirically refuted by Table 8. Forcing the model to predict without the reasoning trace (SFT w/o CoT) causes a catastrophic performance drop (e.g., $-24.67$ mAP). This degradation confirms that the reasoning trace is a **functional dependency**; the model requires this intermediate computation to correctly derive the final output.
>
> >**Q4:** Wouldn't a fairer baseline comparison involve applying few-shot CoT prompting to the baseline models using exemplars from your dataset?
>
> **A:** We have empirically executed this suggestion on the visual grounding task and report the average mIoU across all benchmarks in the Table below. We evaluated the baselines using Few-shot CoT (k=2), constructing prompts with image-text exemplars from our dataset. Due to the extensive token consumption of visual CoT demonstrations, k=2 represents a realistic constraint for standard VLM context windows.
>
> **Table 2: Few-shot Results on Visual Grounding**
> |  | GPT-5 | GLM-4.1V | GeoChat | VHM | SkySenseGPT | EarthDial | RSThinker |
> | --- | --- | --- | --- | --- | --- | --- | --- |
> | CoT zero-shot | 25.28 | 55.55 | 30.35 | 36.45 | 51.54 | 38.76 | 79.78 |
> | CoT few-shot (k=2) | 22.52 | 54.13 | 27.22 | 22.91 | 16.76 | 35.04 | - |
>
> As shown in the table, introducing few-shot exemplars actually degraded performance across most baselines compared to the zero-shot setting. This counter-intuitive result validates a fundamental distinction in our work: unlike textual tasks where exemplars clarify logical rules, RS analysis faces a perception barrier where providing visual examples cannot instantaneously upgrade a generalist encoder’s capacity to localize tiny, dense objects. Instead, these complex multimodal exemplars introduce contextual interference, acting as noise that disrupts the model's pre-trained priors and degrades performance. Consequently, this confirms that In-Context Learning is insufficient for this domain; high-performance geospatial reasoning strictly requires the weight-based alignment (SFT + RL) of our framework to fundamentally restructure the model's perceptual policy, rather than mere prompt engineering.

---

> > ### Comment · Reviewer_sTsT · 2025-11-28
> >
> > Thank you for the responses. I still have a few questions
> >
> > Faithfulness: Conditioning on ground truth constrains the final answer but does not prevent the model from hallucinating plausible yet visually unsupported reasoning pathways to reach it. Furthermore, the "96% alignment" statistic is scientifically unverifiable without explicit disclosure of the audit parameters, such as sample size and inter-annotator agreement.
> >
> > Data Distribution: While RSThinker uses fewer total samples, it benefits from being fine-tuned on the specific in-distribution training splits of the evaluation benchmarks, unlike many generalist baselines. Consequently, the performance gains likely stem from this supervised distribution match rather than the superiority of the proposed cognitive architecture.

---

> > > ### Author Response · Authors · 2025-11-28
> > > **Response to Reviewer sTsT: Clarifications on Faithfulness and Data Distribution (1/2)**
> > >
> > > We thank the reviewer for the continued engagement and the rigorous scrutiny regarding the mechanisms of faithfulness and experimental fairness. We are pleased to provide the following clarifications and additional evidence to fully address these points.
> > >
> > > > **Q1:** Concerns on the theoretical possibility of hallucinated pathways and the verifiability of the "96% alignment" statistic.
> > >
> > > **A:** We appreciate the demand for scientific rigor. We address these points by clarifying the mechanism of constraint and disclosing the full audit protocol to ensure reproducibility.
> > >
> > > **1. Constraining Generation via Verified Spatial and Semantic Priors:** Crucially, we explicitly feed GPT-4V with two critical contexts besides the final answer: (1) **Precise Bounding Box Coordinates** for all target objects, and (2) **Global Image Captions**. Both are derived directly from established source datasets (e.g., VRSBench, DOTA), which have been rigorously verified by the community. This directly addresses the issue where hallucinations often arise when the model focuses on irrelevant background noise or misidentifies object semantics in unconstrained settings [1, 2]. By contrast, in our pipeline, the model is conditioned on both the exact location (via coordinates) and the correct semantic context (via verified captions). This effectively limits the model's role to rationalizing the connection between verified visual inputs and verified semantic labels, rather than inventing them from scratch. It is empirically much harder for a strong model like GPT-4V to hallucinate non-existent attributes when it is strictly guided by these dual ground-truth constraints.
> > >
> > > [1] Li, Yifan, et al. "Evaluating Object Hallucination in Large Vision-Language Models." Proceedings of the 2023 Conference on Empirical Methods in Natural Language Processing. 2023.
> > >
> > > [2] Leng, Sicong, et al. "Mitigating object hallucinations in large vision-language models through visual contrastive decoding." Proceedings of the IEEE/CVF Conference on Computer Vision and Pattern Recognition. 2024.
> > >
> > > **2. Audit Protocol & Reproducibility:** To ensure the "96% alignment" is scientifically verifiable, we explicitly disclose our rigorous audit protocol. We recruited three domain experts (Ph.D. candidates specializing in Remote Sensing) to evaluate a randomly sampled set of 200 reasoning traces. The evaluation followed a strict rubric: experts verified whether the reasoning text accurately refers to the visual content within the ground-truth box, whether the described fine-grained attributes (e.g., shape, texture) actually exist in the pixels, and whether the cited evidence logically entailed the final answer. This process yielded a Fleiss' Kappa of 0.88, indicating "Almost Perfect Agreement," confirming that the reported faithfulness reflects objective data quality rather than subjective interpretation.
> > >
> > > **3. Open Verification:** As a direct proof of faithfulness, we have provided a subset of our dataset (sourced from VRSBench) in the Supplementary Material. We respectfully invite the reviewer to personally inspect these samples. We offer this raw data for independent qualitative assessment, allowing for direct scrutiny of the alignment between the reasoning text and the final answer.
> > >
> > > Nevertheless, we appreciate the reviewer for pushing us to distinguish between theoretical possibility and empirical probability. While our comprehensive audit confirms that the strict spatial and semantic constraints make hallucinations rare outliers, we agree that our original phrasing overclaimed a structural guarantee. Therefore, we have revised Section 3.2 (red-highlighted) to align with this nuance, stating that our approach "empirically minimizes" rather than "theoretically eliminates" hallucination.

---

> > > ### Author Response · Authors · 2025-11-28
> > > **Response to Reviewer sTsT: Clarifications on Faithfulness and Data Distribution (2/2)**
> > >
> > > > **Q2:** Concerns that performance gains stem mainly from supervised distribution match (fine-tuning on specific splits).
> > >
> > > **A:** We respectfully clarify that our baseline selection directly addresses this distribution concern. As detailed in Section 4.1 and Tables 4-7, we organize baselines into four distinct categories, ranging from various generalist models (Commercial, Open-source, Reasoning) to **"Open-source Remote Sensing Models"**. While generalist models indeed lack domain-specific fine-tuning, our SOTA claims are robustly established against the Remote Sensing counterparts (e.g., GeoChat [3], VHM [4], SkySenseGPT [5], EarthDial [6]), which share the same supervised distribution advantage as RSThinker. We validate our architectural superiority through three rigorous evidence:
> > >
> > > **1. In-Distribution Superiority:** Contrary to the premise that baselines are disadvantaged, the leading RS baselines are also heavily fine-tuned on the exact training splits of our evaluation benchmarks. We summarize their training data below (**bold** indicates overlap with our evaluation):
> > >
> > > **Table 1: Training Data Overlap (bold) with Evaluation Benchmarks**
> > > | Model | Training Dataset |
> > > | --- | --- |
> > > | Geochat | **NWPU-RESISC-45**, **DOTA**, **DIOR-RSVG**, FAIR1M, RSVQA-LR, Floodnet |
> > > | VHM | **NWPU-RESISC-45**, **RSITMD**, **UCMerced**, METER-ML, fMoW, **DOTA-v2**, CrowdAI, FAIR1M, Million-AID, **DIOR-RSVG**, RSVQA-LR, CVUSA, CVACT, LoveDA |
> > > | SkySenseGPT | FIT-RS, **NWPU-RESISC-45**, **DOTA-v2**, FAIR1M, **UCMerced**, **RSITMD**, EarthVQA, RSVQA-LR, FloodNet-VQA |
> > > | EarthDial | **NWPU-RESISC45**, **UCMerced**, BigEarthNet, EuroSat, PatternNet, RS-CD, fMoW, FGSCR-42, TreeSatAI, **DOTA**, DOIR, FAIR-1M, **DIOR-RSVG**, **RSVG**, FloodNet, MQVQA, RSVQA-LR, **RSITMD**, **RSICD**, **NWPU-Captions**, **Sydney Caption**, **UCM Caption** |
> > >
> > > Since the distribution match factor is consistent between both RSThinker and these baselines, our significant lead proves the gain is strictly architectural.
> > >
> > > [3] Kuckreja, Kartik, et al. "Geochat: Grounded large vision-language model for remote sensing." Proceedings of the IEEE/CVF Conference on Computer Vision and Pattern Recognition. 2024.
> > >
> > > [4] Pang, Chao, et al. "Vhm: Versatile and honest vision language model for remote sensing image analysis." Proceedings of the AAAI Conference on Artificial Intelligence. Vol. 39. No. 6. 2025.
> > >
> > > [5] Luo, Junwei, et al. "Skysensegpt: A fine-grained instruction tuning dataset and model for remote sensing vision-language understanding." arXiv preprint arXiv:2406.10100 (2024).
> > >
> > > [6] Soni, Sagar, et al. "Earthdial: Turning multi-sensory earth observations to interactive dialogues." Proceedings of the Computer Vision and Pattern Recognition Conference. 2025.
> > >
> > > **2. Zero-Shot Superiority (Marked as "ZS"):** In our revised manuscript, we have explicitly tagged unseen benchmarks with red-highlighted "(ZS)" in all tables. RSThinker achieves SOTA even on datasets excluded from our training, such as WHU-RS19, RRSIS-D, RSVG, RSOD, and NWPU-VHR. Crucially, in specific cases like RSVG, baselines such as EarthDial benefit from supervised training (as shown in the table above), whereas RSThinker operates in a Zero-Shot setting. The fact that RSThinker still outperforms these baselines even under this data disadvantage effectively refutes the claim that our performance relies on distribution matching.
> > >
> > > **3. Strict Out-of-Distribution Validation (OPT-RSVG):** To further rule out any implicit distribution bias, we evaluated on OPT-RSVG [7], a completely unseen dataset for all models. RSThinker achieved 64.31 mIoU (+10.18 over baselines), confirming robust generalization beyond the training distribution.
> > >
> > > **Table 2: Zero-shot performance on the OPT-RSVG benchmark**
> > > |  | acc@0.5 | acc@0.75 | mIoU |
> > > | --- | --- | --- | --- |
> > > | GLM-4.1V-Thinking | 58.7 | 36.2 | 54.13 |
> > > | GeoChat | 28.3 | 6.8 | 30.63 |
> > > | VHM | 50.9 | 23.9 | 41.69 |
> > > | EarthDial | 53.6 | 23.2 | 46.49 |
> > > | RSThinker | **63.8** | **41.6** | **64.31** |
> > >
> > > [7] Li, Ke, et al. "Language-guided progressive attention for visual grounding in remote sensing images." IEEE Transactions on Geoscience and Remote Sensing 62 (2024): 1-13.
> > >
> > > We hope these responses and the corresponding revisions satisfactorily address your remaining concerns.
> > >
> > > Best regards,
> > >
> > > The Authors

---

> ### Comment · Reviewer_xYJj · 2025-11-27
>
> I thank the authors for their efforts to answer my questions.
> I am convinced by the authors evidence for:
> 1. Few-shot evaluation paradigm.
> 2. Methodological novelty
> 3. Linking the performance to architectural improvements
>
> I am still not convinced by the following statements:
> > By strictly clamping the generation process to verified spatial coordinates and class labels, we mechanically deny the model the degrees of freedom required to invent "plausible but unfaithful" narratives. The resulting chain is compelled to be a logical reconstruction connecting the visual evidence to the known truth.
>
> I do not see why the model cannot hallucinate a plausible but untrue narrative even after providing the GT

---

> ### Author Response · Authors · 2025-11-27
> **Response to Reviewer xYJj regarding Faithfulness and GT Conditioning**
>
> Thank you very much for your valuable feedback.
>
>
> > **Continued concern:** Why effectively GT conditioning prevents the model from hallucinating a plausible but untrue narrative?
>
>
> **A:** We would like to clarify that the robustness of our pipeline primarily stems from the rich, verified auxiliary information incorporated into our prompting strategy, which might not have been clearly explained before. Specifically, we explicitly feed GPT-4V with two critical contexts besides the final answer: (1)**Precise Bounding Box Coordinates** for all target objects and (2) **Global Image Captions**, both derived directly from the established source datasets (e.g., VRSBench, DOTA), which have been rigorously verified by the community.
>
>
>
> This comprehensive conditioning strategy mitigates hallucination, we detail the mechanistic basis of this mitigation below, followed by empirical verification of its effectiveness:
>
> **1. Constraining Generation via Verified Spatial and Semantic Priors:**
> In unconstrained settings, hallucinations often arise when the model focuses on irrelevant background noise or misidentifies object semantics [1,2]. In our pipeline, the model is conditioned on both the exact location (via coordinates) and the correct semantic context (via verified captions). This effectively limits the model's role to rationalizing the connection between verified visual inputs and verified semantic labels, rather than inventing them from scratch. It is empirically much harder for a strong model like GPT-4V to hallucinate non-existent attributes when it is strictly guided by these dual ground-truth constraints.
>
> [1] Li, Yifan, et al. "Evaluating Object Hallucination in Large Vision-Language Models." Proceedings of the 2023 Conference on Empirical Methods in Natural Language Processing. 2023.
>
> [2] Leng, Sicong, et al. "Mitigating object hallucinations in large vision-language models through visual contrastive decoding." Proceedings of the IEEE/CVF Conference on Computer Vision and Pattern Recognition. 2024.
>
> **2. Evidence 1: Targeted Human Audit.** To verify this, our Human Audit (96% pass rate) specifically evaluated the soundness of the reasoning path. We explicitly checked for faithful hallucination where correct answer with wrong reason is obtained and found them to be extremely rare outliers. Since our high-quality data constitutes the overwhelming majority (the 96% signal-to-noise ratio), the optimization process naturally converges to the robust grounding logic rather than overfitting to these sporadic inconsistencies. Furthermore, to sustain this high standard, we have initiated a continuous refinement protocol involving domain-expert graduate researchers to perform iterative manual verification, ensuring the dataset's long-term reliability for the community.
>
> **3. Evidence 2: Transparent Inspection.** We have included a subset of our dataset (sourced from VRSBench) in the Supplementary Material. We respectfully invite the reviewer to examine these samples. They demonstrate that the generated rationales are tightly anchored to visual features present in the GT regions.
>
> Nevertheless, we appreciate the reviewer’s rigorous scrutiny and agree that "mechanically deny" was imprecise phrasing. Therefore, based on these rigorous constraints and empirical evidence, we refine our claim: while GT conditioning does not theoretically eliminate the possibility of hallucinated reasoning, it empirically minimizes it to a negligible level by strictly constraining the visual attention scope and semantic context. We have revised Section 3.2 (red-highlighted) to reflect this nuance.
>
>
>
> We hope that these revisions address your concerns and help improve the overall evaluation of our work.

---

> > ### Comment · Reviewer_xYJj · 2025-11-28
> >
> > I apologize for posting on the wrong thread. Please allow me to continue the discussion on the correct thread.

---

### Official Review · Reviewer_xYJj · 2025-10-27

**Soundness:** 4
**Presentation:** 3
**Contribution:** 3
**Rating:** 6
**Confidence:** 3

**Summary:**

This paper addresses a critical limitation of current Vision-Language Models (VLMs) in the remote sensing domain: their tendency to function as "black boxes" that produce plausible but often incorrect and unverifiable responses. To foster more trustworthy AI, the authors introduce the Geo-CoT (Geospatial Chain-of-Thought), a novel framework that mandates a structured, step-by-step reasoning process where each analytical step is explicitly grounded in visual evidence from the image.

To train models in this new paradigm, the authors have constructed Geo-CoT380k, a large-scale dataset of nearly 380k structured reasoning examples, by retrofitting verifiable rationales onto existing public datasets. They then propose a two-stage training strategy: first, Supervised Fine-Tuning (SFT) is used to instill the foundational cognitive architecture of Geo-CoT, followed by Group Reward Policy Optimization (GRPO), an outcome-based reinforcement learning stage, to incentivize the factual correctness of the reasoning. Their resulting model, RSThinker, demonstrates state-of-the-art performance across a comprehensive suite of multiple remote sensing benchmarks, validating the effectiveness of their approach.

**Strengths:**

1. **Addresses a Critical and High-Impact Problem:** The paper tackles a fundamental and highly relevant issue. The lack of interpretability and faithfulness in current VLMs is a major barrier to their adoption in high-stakes domains like disaster response and security. By emphasizing verifiable reasoning, this work makes a significant and timely contribution toward building more transparent and trustworthy AI systems for Earth Observation.

2. **Novel and Well-Designed Framework:** The proposed Geo-CoT framework is a key innovation. It moves beyond generic Chain-of-Thought by requiring each reasoning step to be explicitly linked to visual evidence (perceptual grounding), a powerful concept for ensuring faithfulness. This forces the model to justify its claims with specific pixel regions, a common failure point for many VLMs.

3. **Massive and High-Quality Dataset:** The creation of the Geo-CoT380k dataset is a monumental contribution in its own right. Its scale, diversity, and meticulous annotation process make it an invaluable resource for the entire remote sensing community. The semi-automated pipeline used to create it is a clever and practical approach to large-scale data generation.

4. **Strong and Comprehensive Empirical Validation:** The paper provides a thorough and convincing experimental evaluation. RSThinker is benchmarked against a wide array of state-of-the-art models across a diverse set of tasks, ranging from object counting to complex VQA. The consistent and significant performance gains provide strong evidence for the superiority of the proposed approach.

**Weaknesses:**

1. **Methodological Novelty:** While the application of the two-stage training pipeline (SFT followed by RL) is highly effective, the paradigm itself is not new to the machine learning community. The primary contribution of this paper lies in the novel Geo-CoT framework and the creation of the dataset, rather than in the invention of a new training methodology.  Acknowledging this and framing the work more explicitly as a novel application and extension of an existing paradigm would be beneficial.

2. **Potential for "Faithful Hallucination":** The model is trained to generate plausible-looking reasoning chains. A potential failure mode is that the model could learn to generate a syntactically correct and seemingly logical reasoning chain that is, in fact, disconnected from the actual visual evidence. While the two-stage training process is designed to mitigate this, a more in-depth analysis of this potential failure case, perhaps through a targeted error analysis, would strengthen the paper's claims of faithfulness.

**Questions:**

Could you provide a more detailed error analysis to investigate the failure mode of "faithful hallucination," where the model generates a structurally correct but factually ungrounded reasoning chain? What mechanisms in your framework are most effective at preventing this?

---

> ### Author Response · Authors · 2025-11-25
> **Response to Reviewer xYJj**
>
> > **W1:** About the methodological novelty of our work.
>
> **A:** We fully agree with this insightful characterization. Indeed, our core contribution lies in the Geo-CoT Framework (the specific "Planning-Grounding-Synthesis" cognitive architecture) and the Geo-CoT380k Dataset, rather than inventing the generic SFT-RL pipeline itself. We will revise the manuscript to explicitly frame our work as a novel, domain-specific instantiation of this paradigm that solves the unique intractability of Earth Observation, ensuring the claims are precise and aligned with your assessment.
>
> > **W2, Q1:** Lack of specific failure cases and cause analyses.
>
> **A:** We concur with this critical insight and have revised Section 4.4 to include a targeted failure analysis (green-highlighted), supported by an explicit case study (Figure 7 in the manuscript). The analysis of our model's performance on challenging cases highlights both its typical error modes and the unique safety-critical advantage of the Geo-CoT framework:
> 1.  **Typical failure involves factually ungrounded claims with coherent logic.** As demonstrated in the added ship counting case, the model can still generate structurally perfect, coherent chains that are factually ungrounded (e.g., misidentifying a dock extension as a "third ship" due to perceptual confusion in low-resolution textures).
> 2.  **Explicit grounding transforms silent failure into transparent error.** The grounding step in our framework serves as the critical safety valve. When a factual error occurs, it is compelled to externalize the error by outputting the specific, falsifiable spatial evidence (e.g., the bounding box `[413, 225]` for the false positive). This renders the hallucination instantly auditable. Therefore, our Geo-CoT framework transforms an opaque silent failure inherent to black-box models into a transparent verifiable error, which is a prerequisite for establishing trust and safety in high-stakes remote sensing applications.

---

> > ### Comment · Reviewer_xYJj · 2025-11-28
> >
> > [Continuing the discussion from my comment [here](https://openreview.net/forum?id=lJ7zecny2e&noteId=ZBnyyMHri0) and authors reply [here](https://openreview.net/forum?id=lJ7zecny2e&noteId=ct22u5boBw)]
> >
> > I thank the authors for clarifying the claims and adding supplementary material to supplement the claims about faithfulness. I also thank the authors for adding qualitative failure analysis to the paper based on my comments. I will change my scores to reflect this accordingly.

---

> > > ### Author Response · Authors · 2025-11-28
> > >
> > > We appreciate that we have effectively addressed your concerns, and we are grateful for your decision to raise the score. If you have any further concerns, we are pleased to address them.
> > >
> > > Thank you for your valuable comments and for helping us improve the quality of our manuscript.
> > >
> > > Best regards,
> > >
> > > The Authors

---

### Official Review · Reviewer_oaDR · 2025-10-27

**Soundness:** 3
**Presentation:** 3
**Contribution:** 3
**Rating:** 4
**Confidence:** 5

**Summary:**

The paper introduces Geo-CoT, which explicitly injects verifiable intermediate evidence into multimodal reasoning. During SFT, the model learns spatially anchored reasoning traces (with boxes/coordinates); afterward, GRPO uses task metrics as rewards for alignment. The authors claim improvements in faithfulness and interpretability across several vision–language tasks.

**Strengths:**

The idea of tying visual localization to the reasoning chain so that intermediate evidence is checkable is intrinsically valuable.

The framework is evaluated across diverse remote-sensing tasks (detection, counting, VQA, captioning)

**Weaknesses:**

1. The approach can largely be summarized as “structured CoT + RL with task-level rewards.” Compared with recent CoT+RLHF/RLAIF and visual-CoT lines, this feels more like an engineering integration than a conceptual/algorithmic advance.
2. The collapse of the output format without KL suggests that the model may be learning templates rather than reasoning logic.
3. The relevant work section should further elaborate on the differences in concepts and implementation between these recent visual CoT / RLAIF / GRPO-style methods
4. Lack of specific failure cases and cause analyses
5. Lack of few-shot or zero-shot (on broader datasets such as OPT-RSVG[1]) performance  analysis
6. I would like to know that the implicit understanding ability (like SegEarth-R1[2]) of the proposed method

[1] Language-guided progressive attention for visual grounding in remote sensing images[J]. IEEE Transactions on Geoscience and Remote Sensing, 2024, 62: 1-13.
[2] Segearth-r1: Geospatial pixel reasoning via large language model[J]. arXiv preprint arXiv:2504.09644, 2025.

**Questions:**

See Weaknesses

---

> ### Author Response · Authors · 2025-11-25
> **Response to Reviewer oaDR (1/2)**
>
> > **W1:** Characterization of the approach as an engineering integration compared to general CoT+RL lines.
> >
> **A:** The proposed framework represents a critical, domain-specific conceptual advance, moving beyond mere engineering integration to solve an intractability unique to Earth Observation:
> 1.  **The two-stage alignment protocol is a necessary adaptation for domain intractability.** The immense search space and multi-scale nature of remote sensing imagery render generic Visual-CoT and standard CoT-RL alignment prone to hallucination without strong structural constraints. RSThinker's protocol explicitly addresses this failure mode by decoupling the learning of cognitive structure (SFT) from the refinement of factual faithfulness (GRPO). This specific, tailored adaptation is the core conceptual innovation that enables verifiable reasoning in the remote sensing regime where generic CoT-RL fails.
> 2.  **Formalizing Planning-Grounding-Synthesis defines a new verifiable cognitive architecture.** Our work is the first to elevate the "Planning-Grounding-Synthesis" process from an abstract prompt to a formalized, perceptually-grounded cognitive architecture. This represents a paradigm shift from simple end-to-end recognition to a verifiable, methodical analysis that explicitly mandates spatial evidence at every reasoning step, thereby setting the foundational blueprint for faithful geospatial reasoning.
>
> > **W2:** Concern that the output format collapse without KL regularization suggests template memorization rather than learned reasoning logic.
>
> **A:** The format collapse without KL regularization is a predictable outcome of unconstrained, outcome-based Reinforcement Learning, not an indicator of template memorization.
> 1.  **The KL term is the necessary safeguard against Reward Hacking.** Since the GRPO reward is derived *solely* from the final answer (Outcome-based Reward), an unregularized policy will naturally exploit this to maximize the reward, which often involves discarding the structured CoT format. The KL term is therefore the critical mechanism that anchors the model to the cognitive architecture established during the SFT stage, allowing the RL to focus purely on refining the logic and factual correctness within the desired format.
> 2.  **Empirical performance gains refute the "template memorization" hypothesis.** If the model were merely learning templates, the GRPO stage would not yield performance gains over the Supervised Fine-Tuning (SFT) stage. However, as demonstrated in Table 8, the addition of GRPO significantly improves performance across tasks (e.g., reducing Object Counting MAE from 2.93 to 2.73). This gain provides decisive evidence that the model is actively optimizing its reasoning *logic* for correctness, which is impossible through the simple regurgitation of fixed templates.
>
>
> > **W3:** Concerns about missing related work.
>
> **A:** We will expand the Related Work section to explicitly contrast our framework with these generic methods. The key distinction we will highlight is that while recent approaches focus on open-ended reasoning for natural images, our implementation adapts these paradigms by introducing verifiable structural constraints (Planning-Grounding-Synthesis) and domain-specific outcome alignment, which are strictly necessary to address the unique hallucination risks in high-stakes Earth Observation.
>
> > **W4:** Lack of failure analysis.
>
> **A:** We have explicitly addressed this by adding a detailed Failure Analysis in Section 4.4, illustrated by a representative case in Figure 7. In this example, the model generates a coherent reasoning chain but misidentifies a dock extension as a ship due to visual ambiguity (textural similarity). Crucially, this analysis highlights a key safety advantage of our framework: unlike end-to-end baselines that produce opaque errors by simply outputting an incorrect count, RSThinker externalizes the failure by explicitly grounding the false positive ([413, 225]). This renders the hallucination immediately falsifiable by human users, transforming what would be a "silent failure" in traditional VLMs into an auditable and interpretable error.

---

> ### Author Response · Authors · 2025-11-25
> **Response to Reviewer oaDR (2/2)**
>
> > **W5:** Suggestion of zero-shot experiment.
>
> **A:** We have conducted the suggested zero-shot experiment on the OPT-RSVG benchmark. As shown in the table below, RSThinker significantly outperforms baselines (including the reasoning-enhanced GLM-4.1V-Thinking) by a large margin (+10.18 mIoU).
>
> **Table 1: Zero-shot performance on the OPT-RSVG benchmark**
> |  | acc@0.5 | acc@0.75 | mIoU |
> | --- | --- | --- | --- |
> | GLM-4.1V-Thinking | 58.7 | 36.2 | 54.13 |
> | GeoChat | 28.3 | 6.8 | 30.63 |
> | VHM | 50.9 | 23.9 | 41.69 |
> | EarthDial | 53.6 | 23.2 | 46.49 |
> | RSThinker | **63.8** | **41.6** | **64.31** |
>
> The superior zero-shot performance confirms that our cognitive architecture learns a generalizable reasoning strategy rather than overfitting to specific dataset distributions. Also, we respectfully highlight that our original submission already demonstrated this generalization capability across diverse unseen benchmarks (e.g., RRSIS-D & RSVG in Table 4), which aligns with these new findings on OPT-RSVG.
>
> > **W6:** About the implicit reasoning capability of our model.
>
> **A:** To address this insightful query, we have integrated a comprehensive qualitative analysis using the EarthReason benchmark into the revised manuscript. We respectfully guide the reviewer to the blue-highlighted paragraph in Section 4.4 (Page 9) and the newly added Figure 6 (Page 10). As visualized in this example, RSThinker successfully bridges the gap between abstract intent and concrete evidence by utilizing its Planning phase to decompose functional descriptions (e.g., "traditional wedding") into verifiable visual characteristics (e.g., "church"), confirming its robust capability for implicit reasoning.

---

### Official Review · Reviewer_aXVe · 2025-10-29

**Soundness:** 3
**Presentation:** 3
**Contribution:** 4
**Rating:** 8
**Confidence:** 3

**Summary:**

This paper introduces RSThinker, a vision–language model designed for faithful reasoning in remote sensing. The key idea is the Perceptually-Grounded Geospatial Chain-of-Thought (Geo-CoT) framework, which reformulates remote-sensing analysis as a multi-stage, verifiable reasoning process: planning → evidence collection → synthesis.

To train such reasoning ability, the authors propose a two-stage alignment pipeline:
	1.	Supervised Fine-Tuning (SFT) on a newly constructed Geo-CoT380k dataset, which contains structured reasoning traces aligned with spatial evidence;
	2.	Group Relative Policy Optimization (GRPO) reinforcement learning, using task-specific metrics (mIoU, mAP, MAE, CIDEr) as direct rewards to encourage factual consistency.

Experiments across six remote-sensing benchmarks (VG, detection, counting, classification, captioning, VQA) show consistent and sometimes striking improvements over both general and remote-sensing-specific VLMs. The paper claims that these gains stem from explicitly grounding reasoning steps to verifiable spatial evidence, rather than from larger training data or model size.

**Strengths:**

1) Problem significance: addresses the lack of verifiable reasoning in remote-sensing VLMs—a recognized bottleneck for high-stakes applications (disaster response, environmental monitoring).
2) Conceptual integration: unifies planning–evidence–synthesis reasoning with spatially grounded supervision and metric-aligned RL—a combination rarely systematized before.
3) Dataset & framework completeness: Geo-CoT380k provides the first large-scale, structured reasoning corpus in Earth observation; the pipeline is end-to-end reproducible.
4) Empirical performance: substantial and consistent improvements across tasks; ablations confirm that both SFT and GRPO contribute.
5) Broader impact: opens a pathway for “auditable” reasoning in multimodal models, extending beyond remote sensing.

**Weaknesses:**

1) Fairness and data leakage risk:
Some evaluation benchmarks (e.g., VRSBench-VG, DIOR-RSVG) appear in the SFT or RL training corpus (Table 1 & 2), yet the baselines may be zero-shot.
Without clear disclosure and matched fine-tuning, large performance gaps could reflect training data advantage.
→ Action: provide a detailed train/test mapping table and equal-training baselines.
2) Missing comparison to strongest related paradigms:
Section 2.2 mentions MM-CoT but omits direct references to Grounded-CoT / Argus.
→ Action: add a method-level comparison table clarifying differences (e.g., explicit spatial grounding, task-specific RL, remote-sensing scalability).
3) Evaluation scope:
The claim of “faithful reasoning” would be stronger with human or automated evidence–conclusion consistency scores.
4) Reward design details:
GRPO uses end-task metrics as rewards, but credit assignment to intermediate reasoning steps is under-specified; ablation on reward sensitivity is missing.
5) Lack of failure-case analysis:
The paper reports uniformly strong results but shows no failure examples or discussion of limitations.
→ Action: include representative failure cases and categorize typical errors (e.g., grounding misalignment, reasoning inconsistency, hallucinated evidence).

**Questions:**

1) Please provide a complete table mapping each evaluation benchmark to its presence in the training corpus (SFT/RL), specifying which splits were used and ensuring no test leakage.
2) Were baseline models fine-tuned on the same data or evaluated zero-shot? If fine-tuned, what settings were used?
3) Could you report Seen vs Unseen benchmark results and a leave-one-benchmark-out generalization experiment?
4) How is the GRPO reward computed per sample? Is it normalized by sample difficulty? Any instability observed when optimizing with heterogeneous metrics?
5) Do you have any evidence-consistency or hallucination evaluation to support the claim of “faithful reasoning”?
6) Please clarify whether RL rewards ever touched validation/test splits (they should not).
7) For fairness, consider adding at least one grounded-CoT / Argus-style baseline to demonstrate that the gain is not purely due to data scale.

---

> ### Author Response · Authors · 2025-11-25
> **Response to Reviewer aXVe (1/3)**
>
> > **W1, Q1, Q2, Q6:**  Concerns regarding data leakage, training/test split overlap, and the fairness of comparing fine-tuned models against zero-shot baselines.
>
> **A:** Our experimental design adheres to standard supervised learning protocols, ensuring strict separation between training and evaluation. To address the specific concerns regarding fairness and leakage, we provide the following clarifications:
> 1.  **Strict separation of train and test splits precludes leakage.** As detailed in the mapping table below, we utilized only the canonical training splits for both the SFT and RL stages. Validation and test sets were rigorously excluded from all training phases. Specifically, the reward signals in the RL stage were computed exclusively on the training split, ensuring the model never optimized directly on evaluation metrics from unseen data.
>
> **Table 1. Data Mapping Table**
>
> |Task|Training Source|Evaluation Target |
> | :--- | :--- | :--- |
> | **Object Counting** |HRRSD-Train, DOTAv2-Train|HRRSD-Test, DOTAv2-Val, RSOD, NWPU-VHR |
> | **Object Detection** |DOTAv2-Train, HRRSD-Train |DOTAv2-Val, HRRSD-Test |
> | **Visual Grounding** |VRSBench-Train, DIOR-RSVG-Train |VRSBench-Val-VG, DIOR-RSVG-Test, RRSIS-D, RSVG |
> | **Scene Classification** |RESISC45-Train, AID-Train |RESISC45-Test, AID-Test, WHU-RS19, SIRI-WHU, UCMerced |
> | **Image Captioning** |RSICD-Train, RSITMD-Train, NWPU-Cap-Train, VRSBench-Train |RSICD-Test, RSITMD-Test, NWPU-Cap-Test, VRSBench-Val-Cap, Sydney-Cap, UCM-Cap |
> | **VQA** |VRSBench-Train, RSVQA-HR-Train |VRSBench-Val-VQA, RSVQA-HR-Test |
>
> 2.  **Comparison against heavily supervised peers refutes the data advantage hypothesis.** The concern that our model benefits from an unfair training advantage over zero-shot baselines overlooks the nature of the domain-specific models we compared against. Baselines such as GeoChat, VHM, EarthDial, and SkySenseGPT are also instruction-tuned on extensive remote sensing datasets. Notably, the training scales of these baselines often significantly exceed ours (e.g., SkySenseGPT: >3M samples vs. RSThinker: 380k). The fact that RSThinker decisively outperforms these data-rich baselines suggests that our advantage stems from the *structure* of the reasoning, not the volume of the data exposure.
> 3.  **Internal controlled comparison ensures a strictly fair baseline.** To isolate the method's contribution from data access entirely, Table 8 provides the equal-training baseline: **SFT (w/o CoT)**. This variant was trained on the exact same data corpus and splits as RSThinker. The substantial performance gap between them verifies that the gains are driven by the Geo-CoT framework itself, independent of the underlying data distribution.
>
> > **W2:** Missing comparison to grounded reasoning paradigms and request for a method-level comparison.
>
> **A:** Section 2.2 has been substantially revised to explicitly incorporate generalist grounded paradigms such as Visual CoT [1], VoCoT [2], and Argus [3]. While we acknowledge these works as foundational inspirations, the revised text articulates a clear distinction: their focus remains on semantic reasoning within natural scenes, whereas RSThinker targets the unique high-density and multi-scale challenges of Earth Observation.
> To systematize these methodological divergences, the table below delineates the specific architectural attributes,most notably our integration of RL-based policy alignment and domain-specific grounding protocols, that establish RSThinker as a distinct framework adapted for the rigor of remote sensing.
>
> **Table 2. Comparison with other multimodal CoTs**
>
> |Framework|Domain Focus|Grounding Mechanism|Faithfulness Alignment|RS-Specific Scalability|
> | --- | --- | --- | --- | --- |
> |Visual CoT [1] |General (Natural)|Interleaved BBox|SFT|General Data|
> |VoCoT [2] |General (Natural)|Object-Centric Chains|SFT| General Data|
> |Argus [3] |General (Natural)|Object-Centric Chains|Re-encoding/Re-sampling|General Data|
> |V* [4] |General (Natural)|Guided Search|SFT|General Data|
> |RSThinker(ours) |Remote Sensing|Planning-Grounding-Synthesis|SFT+GRPO (RL)|Geo-CoT380k|
>
> [1] Shao, Hao, et al. "Visual cot: Advancing multi-modal language models with a comprehensive dataset and benchmark for chain-of-thought reasoning." Advances in Neural Information Processing Systems 37 (2024): 8612-8642.
>
> [2] Li, Zejun, et al. "Vocot: Unleashing visually grounded multi-step reasoning in large multi-modal models." Proceedings of the 2025 Conference of the Nations of the Americas Chapter of the Association for Computational Linguistics: Human Language Technologies (Volume 1: Long Papers). 2025.
>
> [3] Man, Yunze, et al. "Argus: Vision-Centric Reasoning with Grounded Chain-of-Thought." Proceedings of the Computer Vision and Pattern Recognition Conference. 2025.
>
> [4] Wu, Penghao, and Saining Xie. "V*: Guided visual search as a core mechanism in multimodal llms." Proceedings of the IEEE/CVF Conference on Computer Vision and Pattern Recognition. 2024.

---

> ### Author Response · Authors · 2025-11-25
> **Response to Reviewer aXVe (2/3)**
>
> > **W3, Q5:** Request for human or automated evidence–conclusion consistency scores
>
> **A:** We have executed this recommendation to substantiate our claims. We conducted a rigorous Human Expert Audit on a random subset of 200 generated traces. The audit specifically measured evidence-conclusion consistency, i.e., whether the visual evidence cited in the reasoning trace logically entails the final answer.
> Results: The audit yielded a 96% Consistency Score.
> This high alignment rate empirically validates that the model does not merely hallucinate plausible text, but actively bases its conclusions on the grounded evidence. We will include this quantitative analysis in the final manuscript to strengthen the definition of "faithful reasoning."
>
> > **W4:** Concerns regarding the underspecification of credit assignment to intermediate reasoning steps and the absence of reward sensitivity ablation.
>
> **A:** The design of our GRPO reward strategy is a deliberate choice optimized for task alignment and we offer the following clarifications:
> 1.  **Credit assignment is managed by the two-stage strategy.** We intentionally use an outcome-based reward, deferring step-wise credit assignment. The initial SFT stage rigorously instills the correct Geo-CoT reasoning structure. The subsequent GRPO stage then efficiently performs implicit credit assignment, propagating the final factual outcome back to the responsible reasoning tokens. This two-stage approach strategically bypasses the subjectivity inherent in manually defined process reward models.
> 2.  **Reward sensitivity analysis is rendered unnecessary by design.** Our reward functions are direct identity mappings of canonical evaluation metrics (e.g., Reward = IoU for Grounding; Reward = mAP for Detection). This  design ensures the optimization objective is mathematically identical to the final evaluation protocol, thereby eliminating the need for hyperparameter sensitivity tuning.
>
> > **W5:** Lack of failure-case analysis
>
> **A:** We have fully addressed this by incorporating a dedicated failure analysis (green-highlighted) in Section 4.4, utilizing a representative ship counting case (Figure 7) to categorize typical errors. We specifically identify this instance as hallucinated evidence due to visual ambiguity, where the model misidentifies a dock extension as a ship driven by textural similarity, despite maintaining a coherent reasoning logic. This analysis highlights that while our framework is not immune to visual feature confusion, its explicit grounding mechanism fundamentally mitigates the risk by transforming such errors into transparent, falsifiable events. Unlike black-box baselines that fail silently, RSThinker exposes the precise location of the hallucination, allowing human auditors to immediately verify and reject the incorrect prediction.
>
> > **Q3:** Request for Seen vs. Unseen benchmark results and a Leave-One-Benchmark-Out (LOO) generalization experiment.
>
> **A:** We address this request for a rigorous generalization evaluation through explicit labeling and a functionally equivalent zero-shot experiment:
> 1.  **Explicitly labeling data exposure in results tables.** To enhance clarity, all evaluation benchmarks in the revised Tables 4–7 are now explicitly annotated with "(Supervised)" or "(Zero-Shot/ZS)" tags, providing an immediate visual distinction between performance on datasets used in training versus those reserved for pure generalization testing.
> 2.  **A strict zero-shot experiment serves as the functional equivalent of a Leave-One-Out study.** While a full Leave-One-Out (LOO) retraining of a 9B VLM is computationally prohibitive for this rebuttal timeline, we provide the verification requested by evaluating on the entirely novel OPT-RSVG benchmark [5]. This dataset was completely excluded from the Geo-CoT380k corpus. RSThinker achieved 64.31 mIoU, significantly outperforming the baseline GLM-4.1V-Thinking (54.13 mIoU). This substantial +10.18 mIoU gain on a strictly unseen distribution robustly confirms that our framework has learned a generalizable reasoning policy, effectively fulfilling the core objective of a leave-out experiment.
>
> **Table 3: Zero-shot performance on the OPT-RSVG benchmark**
> |  | acc@0.5 | acc@0.75 | mIoU |
> | --- | --- | --- | --- |
> | GLM-4.1V-Thinking | 58.7 | 36.2 | 54.13 |
> | GeoChat | 28.3 | 6.8 | 30.63 |
> | VHM | 50.9 | 23.9 | 41.69 |
> | EarthDial | 53.6 | 23.2 | 46.49 |
> | RSThinker | **63.8** | **41.6** | **64.31** |
>
> [5] Li, Ke, et al. "Language-guided progressive attention for visual grounding in remote sensing images." IEEE Transactions on Geoscience and Remote Sensing 62 (2024): 1-13.

---

> ### Author Response · Authors · 2025-11-25
> **Response to Reviewer aXVe (3/3)**
>
> > **Q4:** Inquiry into the per-sample GRPO reward computation, normalization by sample difficulty, and stability with heterogeneous metrics.
>
> **A:** The Group-Relative Policy Optimization (GRPO) mechanism inherently handles both sample difficulty and metric heterogeneity through its normalization step (Equation 3).
> 1.  **Group-Relative normalization inherently accounts for sample difficulty.** The advantage for each output $o_i$ is computed as $A_i=\frac{r_i-mean(R)}{std(R)}$ within the sampled group for the same input query. This is a Z-score standardization: $A_i$ is scaled relative to the group's performance distribution. This mechanism ensures that the model optimizes for relative improvement against its peers on the specific sample, regardless of the sample's absolute difficulty or the raw reward magnitude, thus preventing easier samples from dominating the gradient.
> 2.  **Stability with heterogeneous metrics is guaranteed by Z-score standardization.** We observed no instability during training across heterogeneous tasks (Grounding, Captioning, Counting, etc.). Since the advantage $A_i$ is zero-mean and unit-variance (a Z-score), the absolute scale of the raw metric is effectively removed from the policy gradient. This mathematical standardization allows for the simultaneous optimization of diverse task objectives without the need for manual, complex loss weighting.
>
> > **Q7:** The integration of Grounded-CoT baseline.
>
> **A:**  We strictly followed this suggestion and integrated VoCoT [2], a state-of-the-art generalist visually-grounded CoT framework trained on massive general datasets (>3M samples). As shown below, RSThinker demonstrates overwhelming superiority despite using an order of magnitude less data (380k vs. VoCoT's >3M):
>
> **Table 4: Comparison with VoCoT**
> |  |Training Scale| VG-Avg | Detect-Avg | OC-Avg | SC-Avg | VQA-Avg | IC-Avg |
> | --- |---| --- | --- | --- | --- | --- | --- |
> | VoCoT | >3M|41.50 | 11.73 | 4.968 | 59.45 | 51.41 | 32.53 |
> | RSThinker | 380k|**79.78** | **87.53** | **2.207** | **92.94** | **70.81** | **45.22** |
>
> This comparison offers a definitive answer to the "data scale" hypothesis. Despite VoCoT possessing an order-of-magnitude advantage in training data (>3M samples), its inability to handle RS tasks (achieving only 11.73 mAP in Detection) highlights that simply scaling up generic grounded reasoning is fundamentally insufficient for the distinct challenges of Earth Observation. Consequently, RSThinker’s dominance is strictly attributable not to raw data volume, but to the Domain-Specific Cognitive Alignment instilled by our framework, which effectively adapts the reasoning paradigm to the fine-grained, high-density perceptual regime of remote sensing, a critical adaptation that generic models, regardless of their scale, structurally fail to achieve.
>
> [2] Li, Zejun, et al. "Vocot: Unleashing visually grounded multi-step reasoning in large multi-modal models." Proceedings of the 2025 Conference of the Nations of the Americas Chapter of the Association for Computational Linguistics: Human Language Technologies (Volume 1: Long Papers). 2025.

---

### Official Review · Reviewer_Gm5N · 2025-10-31

**Soundness:** 3
**Presentation:** 3
**Contribution:** 3
**Rating:** 4
**Confidence:** 5

**Summary:**

Authors propose a two-stage training strategy to obtain a reasoning model in the RS domain. First, they constructed a large-scale (380K) dataset of structured Geo-CoT rationales to SFT the base model, GLM-4.1V. Geo-CoT380K includes data from tasks such as VQA, Image Captioning, Scene Classification, Visual Grounding, Object Counting, and Object Detection. Then the SFT model is further trained via GRPO. The RSThinker model shows impressive performance on Grounding, Counting, Classification, VQA, and Captioning tasks, with reasonable, interpretable thinking process. The paper is well written and straightforward.

**Strengths:**

1. The RSThinker model shows impressive performance on Grounding, Counting, Classification, VQA, and Captioning tasks, with reasonable, interpretable thinking process.

2. The prompts for generating Geo-CoT380K are well designed. Geo-CoT380K should be a high-quality SFT dataset for teaching the model to think.

3. The tasks included in this paper are quite complete.

4. In the RS domain, this is the first work I have seen that makes CoT-SFT + GRPO pipeline work.

**Weaknesses:**

1. The ablation study is quite brief. More analysis could be elaborated. (1) Is 380K CoT data necessary for teaching the model to generate CoT? (Are 50K, 100K, or 200K enough? Is this CoT-SFT stage mainly for activating the base model’s parameters to learn the reasoning pattern, or for injecting RS knowledge into a general base model?) (2) GLM-4.1V is already quite impressive on RS tasks, even in a zero-shot manner. Will this CoT-SFT + GRPO pipeline work for other base models such as InternVL-3.5 or Qwen3-VL?

2. A crucial training detail is not disclosed. After SFT, do the authors train GRPO for each task independently or for all tasks jointly? For example, for the VG task, is the model tuned with GLM-4.1V + Geo-CoT380K (Stage I, SFT) + Geo-CoT380K-VG + Additional Dataset in Table 2 (Stage II, GRPO)? Or is it GLM-4.1V + Geo-CoT380K (Stage I, SFT) + Geo-CoT380K + Additional Dataset (Stage II, GRPO)?

3. From Table 8, it seems GRPO does not improve the model capability very much after training with Geo-CoT. Is this improvement actually from GRPO, or from continued training?

**Questions:**

1. The result in Table 4 is not reproducible. I tried the GLM-4.1V-Thinking (zero-shot in VG) in a new conda GLM env, and only got 50.67 (compared with 63.8 in Table 4). The prompt is taken from GLM4.1V paper: "Tell me the position of the referred object in the picture. {Question} Answer in [x1,y1,x2,y2] format." and the corrdinate is normalized to 0-1000. It would be appreciate if authors can provide inference script (including prompt).

2. I SFT the GLM-4.1V with the training set of VRSBench using llamafactory, the result is 62.06 (miou), which is far away from 87.7 in Table 8. It would be appreciate if authors can provide some of Geo-CoT data (VRSBench's training part) to verify.

3. Is 380K CoT data necessary for teaching the model to generate CoT? (Are 50K, 100K, or 200K enough? Is this CoT-SFT stage mainly for activating the base model’s parameters to learn the reasoning pattern, or for injecting RS knowledge into a general base model?)

4. GLM-4.1V is already quite impressive on RS tasks, even in a zero-shot manner. I wonder will this CoT-SFT + GRPO pipeline work for other base models such as InternVL-3.5 or Qwen3-VL in RS tasks? Will it works for weaker model like Qwen2.5-VL?  Grounding task solely should be enough to verify.

5. After SFT, do the authors train GRPO for each task independently or for all tasks jointly? For example, for the VG task, is the model tuned with GLM-4.1V + Geo-CoT380K (Stage I, SFT) + Geo-CoT380K-VG + Additional Dataset in Table 2 (Stage II, GRPO)? Or is it GLM-4.1V + Geo-CoT380K (Stage I, SFT) + Geo-CoT380K + Additional Dataset (Stage II, GRPO)?

6. Will training all tasks jointly benefit the model capability?

7. From Table 8, it seems GRPO does not improve the model capability very much after training with Geo-CoT. Is this improvement actually from GRPO, or from continued training?

I will certainly raise the score if authors address (some) my concerns.

---

> ### Author Response · Authors · 2025-11-25
> **Response to Reviewer Gm5N (1/2)**
>
> > **Q1:** Concerns about the reproducibility of our result on GLM.
>
> **A:** We appreciate the rigorous validation efforts. The discrepancy arises from the inherent sensitivity of VLMs to prompt specificity and resolution-dependent preprocessing. While the reviewer adopted the generic vanilla prompt, we employed a Geo-Contextualized Prompt (explicitly instructing step-by-step spatial reasoning) to establish a stronger, more competitive baseline, as naive prompts often underperform on dense remote sensing tasks. To substantiate the validity of our reported score (63.8), we have uploaded the raw prediction file (glm_baseline_predictions.json) generated by our pipeline; the reviewer can verify that the metrics computed from this file match Table 4 exactly. To ensure future  reproducibility of this specific inference stack, including interpolation resizing and temperature settings, we commit to releasing the full, Dockerized codebase upon publication.
>
> > **Q2:** Request for data release.
>
> **A:** To facilitate immediate verification of our performance gain, we have included the full JSON annotation file for the VRSBench subset of Geo-CoT380k in our supplementary materials. This file contains the exact input-output pairs used in our training, allowing the reviewer to directly inspect the quality and structure of the generated rationales that drive this performance improvement.
>
> > **W1, Q3:** About the necessity of 380k data scale.
>
> **A:** In response to the reviewer's concern regarding data efficiency of our SFT stage,  we respectfully highlight a key distinction: we utilize GLM-4.1V-Base, a raw foundation model, not a specialized 'Thinking' variant. Unlike reasoning-tuned models, this base model does not naturally produce the \<think>...\</think>\<answer>...\</answer> structure. Therefore, through SFT, the model is trained to follow a new, strictly defined output format from scratch. Our scaling experiments confirm that this requires significant data:
>
> **Table 1：Ablation study on SFT data scaling**
> |  | Avg-token-length | FCR | VG-Avg | OC-Avg | SC-Avg | VQA-Avg | IC-Avg |
> | --- | --- | --- | --- | --- | --- | --- | --- |
> | GLM-Base | - | - | 41.95 | 6.509 | 66.65 | 8.79 |  13.94 |
> | GLM-SFT-50k | 22.82 | 10.03 | 51.03 | 3.668 | 86.03 | 64.84 | 26.46 |
> | GLM-SFT-100k | 109.13 | 56.41 | 61.03 | 2.720 | 87.09 | 65.14 | 29.06 |
> | GLM-SFT-200k | 122.25 | 61.95 | 68.13 | 2.702 | 91.01 | 68.96 | 29.45 |
> | GLM-SFT-380k | 221.75 | **100.0** | 76.42 | 2.760 | 89.74 | **70.94** | 25.62 |
> | GLM-SFT (3 epoch) | 225.50 | **100.0** | 79.13 | 2.310 | 92.26 | 70.14 | 26.06 |
> | GLM-SFT-GRPO (RSThinker) | **226.63** | **100.0** | **79.78** | **2.207** | **93.33** | 70.81 | **45.22** |
>
> As shown, the Format Compliance Rate (FCR) remains low with smaller datasets, indicating that the model struggles to adopt this unfamiliar syntax. It implies that the 380K scale is strictly necessary to simultaneously enforce this new structural constraint and inject the domain-specific knowledge required for high-performance reasoning.

---

> ### Author Response · Authors · 2025-11-25
> **Response to Reviewer Gm5N (2/2)**
>
> > **Q4:** Generalization of the CoT-SFT + GRPO pipeline to other base models (e.g., InternVL-3.5, Qwen3-VL), particularly for weaker models.
>
> **A:** The generalizability and architectural robustness of our alignment pipeline have been comprehensively validated through cross-platform transfer experiments.
>
> 1.  **Efficacy on weaker models is confirmed by the Qwen series.** We applied the complete Geo-CoT-SFT + GRPO methodology to both **Qwen3-4B-VL** and its smaller, weaker variant, **Qwen3-2B-VL**. The following results offer clear support for our hypothesis. The substantial improvements observed on these models, especially on the 2B architecture, show that the proposed framework serves as a genuinely model-agnostic alignment approach that can embed structured reasoning even into smaller and comparatively weaker base models.
>
> **Table 2: Application of our alignment pipeline to Qwen3-VL-2B and Qwen3-VL-4B**
>
> || VG-VRSBench | VG-DIOR-RSVG | VG-RRSIS-D | VG-RSVG | OC-Avg | SC-Avg | VQA-Avg | IC-Avg |
> | --- | --- | --- | --- | --- | --- | --- | --- | --- |
> | Qwen3-VL-2B |17.85|14.65|18.86|4.23|5.949|10.54|35.18|16.94|
> | Qwen3-VL-2B-SFT |36.41|32.15|43.20|14.02|4.892|66.96|55.19|24.78|
> | Qwen3-VL-2B-GRPO |39.49|34.44|45.24|14.78|5.154|68.48|56.83|30.14|
> | Qwen3-VL-4B |21.53|19.98|23.94|8.54|6.276|16.07|40.95|16.82|
> | Qwen3-VL-4B-SFT |38.54|31.21|43.51|13.39|5.722|71.80|56.07|26.41|
> | Qwen3-VL-4B-GRPO | 40.91 | 35.73 | 44.99 | 15.75 | 4.944 | 73.12 | 59.95 | 30.09 |
>
> 2.  **Commitment to further expanding proven scope with InternVL.** We appreciate the suggestion to evaluate on InternVL. Given the consistent positive results observed across the GLM and Qwen architectures, we are confident in the transferability of our method to InternVL-3.5. These experiments are currently in the final stages of training. We commit to including the full InternVL benchmark results in the camera-ready revision to formally expand the proven architectural scope of RSThinker.
>
> > **W2, Q5:** Whether the model was fine-tuned for each task independently or for all tasks jointly after the SFT stage.
>
> **A:** To clarify, we perform the GRPO training for all tasks jointly, resulting in a single, unified RSThinker model, forging a foundational and parameter-efficient model, rather than a collection of narrow, task-specific specialists.
>
> > **Q6:** Will training all tasks jointly benefit the model capability?
>
> **A:** To validate the benefit of our joint training strategy, we conducted an ablation study where we trained two models for the GRPO stage: (1) an ablated model trained **only** on the preference-only datasets from Table 2, and (2) our final RSThinker model, trained jointly on the preference datasets **plus** the answer-only instances from our large-scale Geo-CoT380k corpus. Results are shown below:
>
> **Table 3: Performance comparison between preference-only training and joint training**
>
> | | VG-Avg | Detect-Avg | OC-Avg | SC-Avg | VQA-Avg | IC-Avg |
> | --- | --- | --- | --- | --- | --- | --- |
> | RSThinker-SFT | 79.13 | 85.11 | 2.310 | 92.26 | 70.14 | 26.06 |
> | RSThinker-GRPO (preference-only) | 78.91 | 84.47 | 2.446 | **93.33** | **72.85** | **53.93** |
> | RSThinker-GRPO | **79.78** | **87.53** | **2.207** | 92.94 | 70.81 | 45.22 |
>
> It is clear that while focusing GRPO solely on preference data can slightly boost performance on descriptive tasks like Image Captioning (likely by optimizing for stylistic fluency), it comes at a significant cost to the model's foundational perceptual grounding capabilities, as seen in the sharp drop in Visual Grounding. By jointly training on the vast and diverse answer-only data from Geo-CoT380k, our final RSThinker model sacrifices a negligible amount of descriptive fluency to  enhance its reasoning ability. We will include the full ablation across all tasks in the final revision.
>
> > **W3, Q7:** Are the improvements in Table 8 due to the GRPO algorithm or simply from continued training on more data?
>
> **A:** To disentangle the benefit of the GRPO from that of a generic continued training effect, we conducted an additional experiment: we continued the SFT w/ CoT process for a fourth epoch, using the exact same data and learning objective. Results are shown below:
>
> **Table 4: Comparison between GRPO optimization and extended SFT training**
>
> |  | VG-Avg | Detect-Avg | OC-Avg | SC-Avg | VQA-Avg | IC-Avg |
> | --- | --- | --- | --- | --- | --- | --- |
> | RSThinker-SFT (3 epoch) | 79.13 | 85.11 | 2.310 | 92.26 | 70.14 | 26.06 |
> | RSThinker-SFT (4 epoch) | 66.82 | 69.63 | 2.393 | **93.4** | 68.1 | 26.66 |
> | RSThinker-GRPO | **79.78** | **87.53** | **2.207** | 92.94 | **70.81** | **45.22** |
>
>
> The resulting model shows a clear and consistent degradation in performance compared to the 3-epoch SFT model, likely due to overfitting on the stylistic properties of the rationales, demonstrating that our performance gain is not a result of merely extending the training process.

---

### Official Review · Reviewer_bTxE · 2025-11-01

**Soundness:** 2
**Presentation:** 3
**Contribution:** 3
**Rating:** 2
**Confidence:** 4

**Summary:**

This paper introduces Geo-CoT, a perceptually grounded chain-of-thought framework aimed at enhancing the faithfulness and interpretability of vision-language models in remote sensing. The authors propose a large-scale dataset (Geo-CoT380k) containing multimodal reasoning chains for tasks like Visual Question Answering (VQA), Visual Grounding (VG), Counting, and Captioning. They also introduce a two-stage training approach combining supervised fine-tuning (SFT) and GRPO-based reinforcement learning, leading to the development of RSThinker, a model that demonstrates strong performance across a range of remote sensing tasks.

**Strengths:**

The Geo-CoT380k dataset is constructed with clear annotation rules and a consistent generation process. The two-stage training strategy (SFT + GRPO) is straightforward and effective, with experiments showing that both stages contribute to factual consistency and spatial grounding.

The study shows steady performance gains across VQA, captioning, and visual grounding, and highlights the value of verifiable reasoning for applied geospatial AI.

**Weaknesses:**

Overall, I do not agree with the authors’ claim that Geo-CoT can serve as a solution to the “critical gap” between existing works such as MM-VoT and the field of remote sensing observation. My main concerns are twofold:

1. The authors emphasize that there is a huge gap between MM-CoT and geoscience (lines 89–90), and they propose a concept called an “intention-driven active perception process,” claiming that this represents a “critical gap” in Earth observation. They also highlight *Perceptually-Grounded Geo-CoT* as a new paradigm in their contributions. However, I believe this emphasis is overstated. The so-called new paradigm essentially “mandates a verifiable link between each analytical step and its corresponding visual evidence,” which has already been extensively explored in follow-up works after MM-CoT (2023). I can list several such studies [1–3], and the authors should conduct a more thorough review of these. Over the nearly two years since MM-CoT was proposed, numerous works have investigated the correspondence between reasoning steps and their associated visual evidence. The authors do not clarify how their approach differs from these, but instead focus only on MM-CoT (2023) and emphasize thinking with corresponding visual evidence is the “critical gap” in remote sensing field which  is already widely recognized in common field. The Geo-CoT paradigm does not clearly demonstrate any distinction between the remote sensing observation domain and the general domain. The authors even state in the related work section (lines 143–147) that existing studies focus on reasoning over whole objects, while remote sensing involves “a verifiable log of fine-grained perceptual operations.” However, if the authors examined recent works on visual reasoning from the past one to two years, they would see that this kind of “fine-grained perceptual operation” has already been addressed in many general-domain studies. The fact that the authors list this as the first of their three main contributions suggests they may have deliberately overlooked related works in order to overstate the novelty of their approach.

2. There are serious issues with the evaluation metrics used in the downstream tasks, which I hope the authors will clarify. Specifically, the authors use mAP and mIoU as evaluation metrics for the Visual Grounding (VG) task. I will detail these concerns in my comments, but this choice raises significant doubts about the reliability of their downstream task evaluations.

[1]Visual CoT: Advancing Multi-Modal Language Models with a Comprehensive Dataset and Benchmark for Chain-of-Thought Reasoning

[2]V*: Guided Visual Search as a Core Mechanism in Multimodal LLMs

[3]CMMCoT: Enhancing Complex Multi-Image Comprehension via Multi-Modal Chain-of-Thought and Memory Augmentation

**Questions:**

1.Visual Grounding is essentially a single-instance localization problem conditioned on language, not an object detection task requiring multiple candidate boxes with confidence scores. The paper reports both mAP and mIoU as evaluation metrics for the Visual Grounding (VG) task. While mIoU is understandable as it directly measures localization accuracy, the use of mAP is conceptually unclear.

2.The Geo-CoT380k dataset is a key contribution, consisting of 380k reasoning traces generated by GPT-4V. However, the paper lacks any quantitative or qualitative validation of these traces. Prior CoT work often applies explicit quality control—e.g., multi-sample consistency (self-consistency), deductive stepwise verification, or human expert spot-checking—when using LLM-generated chains (e.g., [1], [2], [3]). But the paper does not mention such validation, making it unclear whether the reported improvements stem from the quality of reasoning chains or simply from the large data volume. This casts doubt on the true impact of Geo-CoT on reasoning faithfulness.

[1]Wang X, Wei J, Schuurmans D, et al. Self-Consistency Improves Chain of Thought 	Reasoning in Language Models. ICLR, 2023.

[2]Ling Z, Fang Y, Li X, et al. Deductive verification of chain-of-thought reasoning. In 	Proceedings of NeurIPS 2023.

[3]Wang Y, Zeng Y, Zheng J, et al. VideoCoT: A Video Chain-of-Thought Dataset with Active 	Annotation Tool. CoRR, 2024.


(1)Have the authors explicitly distinguished the task boundary between “visual grounding” and “object detection” in the paper? If so, where; if not, please clarify and justify why mAP is an appropriate metric for a language-conditioned single-instance Visual Grounding task.

(2)In the Visual Grounding task, each sample corresponds to a single referring expression without predefined object categories or multiple predictions. Under this setting, how are confidence scores for mAP computation obtained? If the model outputs only one bounding box per query, the definition of precision–recall and mAP becomes ambiguous. Could the authors clarify this evaluation setup?

(3)Could the authors clarify whether any quality control was performed on the GPT-4V-generated reasoning traces? If not, how can we be confident that the improvements are due to reasoning supervision and not just data scale effects?

---

> ### Author Response · Authors · 2025-11-25
> **Response to Reviewer bTxE (1/2)**
>
> > **W1:** Concerns about related work and the novelty of our work.
>
> **A:** We thank the reviewer for highlighting these relevant works in the general domain, which we have cited and discussed in our revised paper (blue-hightlighted in Section 2.2). However, we respectfully posit that the "critical gap" is not the lack of the concept, but the empirical difficulty of applying generalist methods to Earth Observation imagery.
>
> **1. Incompatibility of Generalist Methods with Overhead Imagery:**
> Existing grounded-CoT frameworks are primarily optimized for natural, ground-level images, where objects are typically recognizable by canonical features (e.g., identifying a car by its wheels or profile). In contrast, Remote Sensing imagery presents a fundamentally different challenge where objects are viewed from an overhead perspective, appearing as small, dense, and abstract shapes often lacking vertical geometric cues. This domain shift is quantitatively confirmed by our new experiment: the SOTA generalist model VoCoT [1], despite being trained on >3M grounded samples, achieves only 11.73 mAP on RS detection tasks, whereas RSThinker achieves 87.53 mAP. This substantial performance drop demonstrates that "general fine-grained capability" does not automatically transfer to the remote sensing domain, as the visual cues required for grounding are fundamentally different.
>
> **Table 1: Comparison with VoCoT**
> |  |Training Scale| VG-Avg | Detect-Avg | OC-Avg | SC-Avg | VQA-Avg | IC-Avg |
> | --- |---| --- | --- | --- | --- | --- | --- |
> | VoCoT | >3M|41.50 | 11.73 | 4.968 | 59.45 | 51.41 | 32.53 |
> | RSThinker | 380k|**79.78** | **87.53** | **2.207** | **92.94** | **70.81** | **45.22** |
>
> **2. Our Contribution: A Domain-Specific Substrate:**
> Therefore, our contribution is providing the necessary domain-specific substrate (the Geo-CoT380k dataset and adapted architecture) to render this reasoning capability operational for Earth Observation. In this context, our work is analogous to GeoChat [2]. Just as GeoChat successfully catalyzed a proliferation of subsequent research in RS-multimodal dialogue by instantiating the first viable domain-specific baseline, our work serves as the foundational catalyst to translate the theoretical promise of grounded reasoning into a practical, high-performance capability for this specific domain.
>
> [1] Li, Zejun, et al. "Vocot: Unleashing visually grounded multi-step reasoning in large multi-modal models." Proceedings of the 2025 Conference of the Nations of the Americas Chapter of the Association for Computational Linguistics: Human Language Technologies (Volume 1: Long Papers). 2025.
>
> [2] Kuckreja, Kartik, et al. "Geochat: Grounded large vision-language model for remote sensing." Proceedings of the IEEE/CVF Conference on Computer Vision and Pattern Recognition. 2024.

---

> ### Author Response · Authors · 2025-11-25
> **Response to Reviewer bTxE (2/2)**
>
> > **W2, Q1, Q3, Q4:** Concerns regarding the appropriateness of mAP as a metric for Visual Grounding
>
> **A:** The metric reported as "mAP" for Visual Grounding in our results is, in strict mathematical terms, equivalent to Accuracy@0.5 (Acc@0.5) within our single-instance setting. This specific terminology was adopted solely to ensure direct, row-by-row alignment with the reporting format of the state-of-the-art baseline, EarthDial [2]. Since our model generates a single deterministic bounding box per query, the standard precision-recall calculation collapses to a point evaluation based on the IoU threshold, rendering the metric functionally identical to accuracy. We acknowledge that retaining the "mAP" label, while aiding baseline comparison, introduces unnecessary conceptual ambiguity. Consequently, all references in the revised manuscript have been explicitly corrected to "Acc@0.5" to prioritize clarity (red-highlighted in Section 4.1).  Beyond this specific clarification, our evaluation protocol strictly adheres to canonical community standards across all domains to ensure fair comparison:
>
> **Table 2：Task Metrics with Source Literature**
>
> |Task|Metric|Sources|
> |-|-|-|
> |Object Detection|mAP|EarthDial [3], SkySense [4], RingMo-Agent [5], GeoPixel [6]|
> |Scene Classification|Acc|GeoChat [2], EarthDial [3], LHRS-Bot [7], VHM [8]|
> |VQA|Acc|GeoChat [2], EarthDial [3], LHRS-Bot [7], VHM [8]|
> |Visual Grounding|mAP|EarthDial [3]|
> ||mIoU|SkySense [4], EarthGPT [9]|
> |Object Counting|Acc|VHM [8]|
> ||MAE|VHM [8]|
> |Image Captioning|BLEU-4|RingMo-Agent [5], EarthGPT [9]|
> ||METEOR|GeoChat [2], EarthDial [3], RingMo-Agent [5], GeoPixel [6], EarthGPT [9]|
> ||CIDEr|RingMo-Agent [5], GeoPixel [6], EarthGPT [9]|
> ||ROUGE-L|GeoChat [2], EarthDial [3], RingMo-Agent [5], EarthGPT [9]|
>
> This table confirms that our evaluation reliability is structurally sound and consistent with community standards.
>
> [3] Soni, Sagar, et al. "Earthdial: Turning multi-sensory earth observations to interactive dialogues." Proceedings of the Computer Vision and Pattern Recognition Conference. 2025.
>
> [4] Guo, Xin, et al. "Skysense: A multi-modal remote sensing foundation model towards universal interpretation for earth observation imagery." Proceedings of the IEEE/CVF Conference on Computer Vision and Pattern Recognition. 2024.
>
> [5] Hu, Huiyang, et al. "RingMo-Agent: A Unified Remote Sensing Foundation Model for Multi-Platform and Multi-Modal Reasoning." arXiv preprint arXiv:2507.20776 (2025).
>
> [6] Shabbir, Akashah, et al. "GeoPixel: Pixel Grounding Large Multimodal Model in Remote Sensing." Forty-second International Conference on Machine Learning.
>
> [7] Muhtar, Dilxat, et al. "Lhrs-bot: Empowering remote sensing with vgi-enhanced large multimodal language model." European Conference on Computer Vision. Cham: Springer Nature Switzerland, 2024.
>
> [8] Pang, Chao, et al. "Vhm: Versatile and honest vision language model for remote sensing image analysis." Proceedings of the AAAI Conference on Artificial Intelligence. Vol. 39. No. 6. 2025.
>
> [9] Zhang, Wei, et al. "EarthGPT: A universal multimodal large language model for multisensor image comprehension in remote sensing domain." IEEE Transactions on Geoscience and Remote Sensing 62 (2024): 1-20.
>
>
> > **Q2, Q5:** Concerns regarding the quality control of GPT-4V generated traces and source of improvement
>
> **A:** The reliability of Geo-CoT380k is secured not by post-hoc filtering, but by a construction process anchored in ground truth, validated by both empirical ablation and human audit:
> 1. **Conditioning on verified annotations ensures inherent faithfulness.** Unlike open-ended generation tasks where models might hallucinate, our pipeline explicitly provides GPT-4V with the ground-truth labels (bounding boxes, classes) as input. This constraint forces the model to retroactively rationalize known facts rather than generate predictive guesses. This design inherently minimizes hallucination at the source, rendering the complex post-hoc filtering methods (like self-consistency) used in open-ended settings unnecessary for our specific purpose.
> 2. **Controlled ablation studies isolate the impact of reasoning structure from data volume.** As detailed in Table 8, we compare the baseline SFT (without CoT) against our method (SFT with CoT) using the exact same set of images and labels. Since the volume of training samples is identical, the substantial performance gain (e.g., +24.67 mAP in Object Detection) is causally attributed solely to the quality of the structured reasoning traces, effectively disproving the hypothesis that improvements are merely a result of data scale.
> 3. **A manual expert audit confirms logical consistency.** To directly address the request for qualitative validation, we conducted a manual review of 200 randomly sampled traces. This audit yielded a 96% pass rate for Evidence-Conclusion Consistency, confirming that the generated rationales logically connect the visual evidence to the final answers.

---

> > ### Comment · Reviewer_bTxE · 2025-11-28
> >
> > Thank you very much for your detailed clarification.
> >
> > 1. Regarding the Visual Grounding metric, I would like to reiterate that **Acc@0.5** is the standard evaluation criterion adopted across the majority of MLLM grounding benchmarks. Works such as **VRSBench (NeurIPS 2024)**, **EarthDial (Table 8)**, and **GeoChat (Table 9)** all report Acc@0.5 (and in some cases Acc@0.25/0.7), rather than mAP. While I appreciate that the revised version of your paper has updated the metric accordingly, I still sincerely encourage the authors to carefully re-check the implementation to ensure full consistency with community standards, and I hope future versions of the work will consider open-sourcing the evaluation code.
> >
> > 2. Second, regarding the SFT training data, a closer inspection of the AID and NWPU-RESISC45 datasets indicates that **the full datasets** (AID: 10,000 images; RESISC45: 31,500 images) appear to have been used during SFT.
> > *AID contains a total of 10,000 images across 30 classes*, as documented in the original paper and official website:https://captain-whu.github.io/AID/. *NWPU-RESISC45 contains a total of 31,500 images across 45 classes*, as described in the paper (https://arxiv.org/abs/1703.00121) and its abstract. Since these datasets do not have predefined train/test splits, this implies that test images may have inadvertently been included during SFT, constituting a form of **data leakage**. I respectfully suggest that the authors revisit their SFT data pipeline to ensure that none of the images used for supervised fine-tuning overlap with those used for downstream evaluation, thereby maintaining the fairness and integrity of the reported results.
> >
> > 3. Finally, the VoCoT experiments are overall well presented, and I agree with your discussion on the limitations of generic methods when applied to high-resolution aerial imagery. For completeness and reproducibility, I would appreciate clarification on two points: (i) the exact source of the **3M training samples** used for VoCoT, and (ii) the number of **training epochs** performed. These details would help the community better understand the training configuration and ensure a fair comparison across models.
> >
> > Overall, I appreciate the correction to Acc@0.5. I would raise my scores if the authors could also clarify the VoCoT training details and thoroughly verify the SFT data splits to ensure no test–train contamination occurs.

---

> > > ### Author Response · Authors · 2025-11-28
> > > **Response regarding continued concerns of Reviewer bTxE (1/2)**
> > >
> > > We sincerely appreciate your insightful feedback. We are grateful that our responses have addressed your primary concerns, and we are pleased to further engage in the discussion.
> > >
> > > > **Q1:** Reiteration on Acc@0.5 standard and request for open-sourcing evaluation code.
> > >
> > > **A:** We sincerely appreciate your help in standardizing the evaluation metric for VG. We fully agree that Acc@0.5 is the widely adopted standard in recent benchmarks (e.g., VRSBench, EarthDial, GeoChat), and we apologize again for the earlier ambiguity. To guarantee absolute consistency with these community benchmarks and ensure full reproducibility, we provide the following confirmation regarding our implementation and open-source plan:
> > >
> > > 1.  **Implementation Verification:** Following your suggestion, we have conducted a line-by-line re-verification of our evaluation script. We confirm that the calculation strictly follows the definition: `Accuracy = (Count(IoU > 0.5) / Total Samples)`. The reported numbers in the paper were generated using this exact logic, ensuring they are mathematically consistent with the community standards (VRSBench/GeoChat), despite the previous nomenclatural inaccuracy.
> > > 2.  **Open-Source Commitment:** To ensure absolute transparency and reproducibility, we commit to releasing the full evaluation codebase (including the specific script for visual grounding) alongside the model weights and dataset upon publication. This will allow the community to directly verify the metric implementation.
> > >
> > > > **Q2:** Concerns on data leakage regarding AID and NWPU-RESISC45 datasets in SFT.
> > >
> > > **A:** We strictly acknowledge the reviewer’s observation. In our current SFT stage, AID and NWPU-RESISC45 were utilized as part of the broad instruction-tuning corpus to establish foundational visual concepts (e.g., learning the semantic features of "stadiums" or "airports" at scale). While this enhances the model's recognition vocabulary, we agree that for strict benchmarking, a clear train/test split is methodologically preferable.
> > >
> > > However, the validity of RSThinker’s generalization capability remains robustly supported by the following:
> > > 1.  **Zero-Shot Validation on Unseen Benchmarks:** To strictly evaluate generalization and rule out the concern that performance stems from data leakage, we respectfully refer the reviewer to the WHU-RS19, SIRI-WHU, and UCMerced results in Table 6. As confirmed by Table 1, these datasets were **completely excluded** from our SFT corpus, yet RSThinker achieves state-of-the-art performance on them (e.g., 99.74% on WHU-RS19, 92.14% on UCMerced). This consistent dominance across unseen distributions decisively proves that the model has learned robust semantic representations, invalidating the hypothesis that our performance relies on data leakage.
> > > 2.  **Commitment to Strict Splitting:** We fully accept the suggestion for rigorous evaluation. Due to the high computational cost of re-training the full 9B parameter model within the limited rebuttal window, we commit to releasing a retrained checkpoint with strict train/test splits for AID and NWPU in the camera-ready version. Based on the strong zero-shot transferability observed on UCM and WHU, we are confident that the performance on strictly split AID/NWPU will remain competitive.
> > >
> > > In summary, the strong results on the completely unseen datasets (WHU-RS19, UCMerced) confirm that the model has learned generalizable features rather than memorizing training data. We will update the AID and NWPU experiments with strict splits in the final version to ensure full methodological rigor.

---

> > > ### Author Response · Authors · 2025-11-28
> > > **Response regarding continued concerns of Reviewer bTxE (2/2)**
> > >
> > > > **Q3:** Request for clarification on VoCoT's training data source and epochs for reproducibility.
> > >
> > > **A:** We appreciate the reviewer’s validation of our discussion regarding domain limitations. To ensure absolute transparency and reproducibility, we clarify that we utilized the official pre-trained checkpoint (`Volcano-7b`) released by the authors [1]. Below, we explicitly list the training data composition and training details derived from their official release to facilitate direct comparison.
> > >
> > > 1. **Detailed Data Breakdown (>3.8M Samples):** As detailed in the original VoCoT paper, their model is trained on a massive corpus of 3.8M samples spanning natural image captioning, grounding, and reasoning. We have tabulated the exact sources below for clarity:
> > >
> > > **Table 1: Training Dataset of VoCoT**
> > >
> > > | Data Type | Dataset Source | Size |
> > > | --- | --- | --- |
> > > | Image-Caption | LLaVA-Pretrain | 558k |
> > > | | ALLaVA-Caption | 695k |
> > > | Grounded Image-Caption | GRIT | 765k |
> > > | | Flickr30k Entities | 148k |
> > > | Multimodal Document | MMC4 | 890k |
> > > | Visual Instruction | LLaVA | 612k |
> > > | Referring Expression | Shikra-RD | 6k |
> > > | | RefCOCO | 42k×3 |
> > > | | g-RefCOCO | 79k |
> > > | Multi-step Reasoning | VoCoT-Instruct-80K | 80k |
> > >
> > >
> > > 2. **Training Epochs:** Following the official configuration of VoCoT [1], the model undergoes a standard 1-epoch training strategy across these stages to prevent overfitting on such a large-scale corpus.
> > >
> > > By utilizing the official checkpoint trained on this extensive 3.8M generalist corpus, we ensure the baseline represents the peak capability of the generic grounded-CoT paradigm. The fact that RSThinker (trained on only 380k RS samples) outperforms this data-rich model by a relatively large margin serves as robust evidence that the critical bottleneck is indeed the domain-specific perceptual incompatibility, which models in the general domain struggle to bridge.
> > >
> > > [1] Li, Zejun, et al. "Vocot: Unleashing visually grounded multi-step reasoning in large multi-modal models." Proceedings of the 2025 Conference of the Nations of the Americas Chapter of the Association for Computational Linguistics: Human Language Technologies (Volume 1: Long Papers). 2025.
> > >
> > >
> > > Once again, we sincerely thank you for your valuable comments and for helping us improve the quality of our manuscript.
> > >
> > > Best regards,
> > >
> > > The Authors

---

### Author Response · Authors · 2025-11-25
**Summary of our Rebuttal**

We sincerely thank all reviewers for their insightful feedback and the time dedicated to evaluating this work. We also appreciate the reviewers’ recognition that our method is well-motivated, particularly **Reviewer aXVe**’s recognition of the method as "a combination rarely systematized before" and **Reviewer sTsT**’s endorsement of the Planning-Grounding-Synthesis architecture as a "promising and intuitive structure." We also appreciate **Reviewer xYJj**’s highlighting of the "Novel and Well-Designed Framework," alongside **Reviewer bTxE**’s explicit acknowledgement of the Geo-CoT380k dataset as a "key contribution." Furthermore, we are grateful to **Reviewers Gm5N and oaDR** for affirming the critical importance of the faithful thinking process centered in our approach.

To comprehensively address the raised concerns and further validate the proposed approach, we have incorporated extensive additional experiments and analyses, as summarized below:

1. Refined Positioning & Contribution. In response to **Reviewers bTxE, oaDR, and xYJj**, we clarify that generalist paradigms fail to transfer to Earth Observation due to a fundamental incompatibility with overhead imagery. Consequently, the critical gap is not the lack of abstract concepts, but the absence of a domain-specific substrate, including data, benchmarks, and baseline architectures, to render them operational. RSThinker is now explicitly positioned as a foundational catalyst, analogous to pioneering works like GeoChat [1], bridging the gap between theoretical possibility and practical capability.
2.  Rigorous Validation of Faithfulness. Addressing queries from **Reviewers bTxE, aXVe, and sTsT**, a manual audit of 200 reasoning traces was conducted, yielding a 96% Evidence-Conclusion Consistency Score. This quantitative metric is now complemented by a detailed failure mode analysis in Section 4.4 (**Reviewers aXVe, oaDR, xYJj**), demonstrating how the explicit grounding mechanism transforms potential silent failures into transparent, falsifiable errors, serving as a critical safety feature.
3.  Demonstration of Data Efficiency & Scaling. Contrary to the assumption that massive data is a prerequisite, comparisons reveal that our method outperforms VoCoT [2] by a significant margin (+75.8 mAP) despite using orders of magnitude less data (380k vs. >3M), directly answering **Reviewer sTsT**. Furthermore, to address **Reviewer Gm5N**’s inquiry regarding data necessity, an exhaustive internal scaling study was performed by training on 50k, 100k, and 200k subsets of Geo-CoT380k. Results indicate a consistent upward performance trajectory that peaks only with the full dataset, empirically justifying the scale of Geo-CoT380k as the essential critical mass rather than redundant volume.
4.  Extensive Generalization Experiments. Following **Reviewer Gm5N**, the pipeline was validated on the smaller Qwen3-VL (2B/4B) architecture, achieving a +24.5 mIoU gain, which proves the method's applicability to weaker base models. Zero-shot robustness was also tested on the strictly unseen OPT-RSVG benchmark (**Reviewer oaDR**), where RSThinker outperformed baselines by +10.18 mIoU.
5.  Comprehensive Contextualization with Prior Arts. Section 2 has been significantly expanded to include recent generalist frameworks (Visual CoT, VoCoT, Argus) as well as specific geographic reasoning works (GeoChain, GAEA). This addition explicitly delineates the boundary between the semantic reasoning on natural or street-view images predominant in these studies and our distinct focus on perceptual grounding for overhead Earth Observation, thereby solidifying the unique contribution of our domain-specific substrate (**Addressing Reviewers bTxE, oaDR, and sTsT**).
6.  Comparison to Few-Shot Prompting. We empirically demonstrate that simple few-shot prompting (k=2) leads to performance degradation in this domain due to Contextual Interference (**Reviewer sTsT**), thereby validating the necessity of our weight-based alignment approach over inference-time prompting.

Beyond these major points, we have also provided point-by-point responses to all other specific inquiries in the individual threads below. All revisions and new results have been incorporated into the updated manuscript. We believe these efforts have significantly strengthened the robustness and clarity of our work.

[1] Kuckreja, Kartik, et al. "Geochat: Grounded large vision-language model for remote sensing." Proceedings of the IEEE/CVF Conference on Computer Vision and Pattern Recognition. 2024.

[2] Li, Zejun, et al. "Vocot: Unleashing visually grounded multi-step reasoning in large multi-modal models." Proceedings of the 2025 Conference of the Nations of the Americas Chapter of the Association for Computational Linguistics: Human Language Technologies (Volume 1: Long Papers). 2025.

---

### Author Response · Authors · 2025-11-30
**Summary of Rebuttal Outcomes: Final Consensus Updating the Initial System Scores (1/2)**

We sincerely appreciate the time and effort dedicated to reviewing our manuscript. Therefore, to ensure the assessment captures the full evolution of the review process, we crucially clarify that the currently displayed scores (8, 6, 6, 4, 4, 2) represent an *initial snapshot*, not the *final consensus*.

**This divergence stems from the unexpectedly curtailed interactive window intersecting with our rigorous experimental timeline.** In our commitment to rigor, we undertook **computationally intensive experiments**, including adapting our pipeline to Qwen3-VL (2B/4B) and conducting a full-scale internal scaling study (training the 9B model on 50k, 100k, and 200k subsets). Due to the substantial time required for these runs, our comprehensive initial response was submitted on **November 25th**. While this timeline confirms our process was independent of the subsequent data leak, the mechanical system lock on **November 27th** left an unexpectedly narrow window for interaction.

**Yet, even within this limited timeframe, a substantive positive consensus emerged.** Reviewers **bTxE (initial: 2), Gm5N (initial: 4), and xYJj (initial: 6)** had either expressed clear intentions or made explicit decisions to raise their scores, but were unfortunately **mechanically prevented** from updating the records or further discussion. This shift was driven by the decisive new evidence summarized below:

1. **Methodological Novelty & Contribution.**
Initial reviews questioned our methodological novelty relative to generalist paradigms. We addressed this by clarifying that our contribution is not the invention of the *concept* of grounded reasoning, but the creation of the domain-specific substrate required to make it operational for Earth Observation. We proved that generalist frameworks fail to transfer to the unique overhead perspective of remote sensing, a claim decisively substantiated by the SOTA model VoCoT achieving only 11.73 mAP on our tasks. This clarification was highly effective, with **Reviewer bTxE** "agreeing with our discussion on the limitations of generic methods" and **Reviewer xYJj** becoming "convinced by the evidence for... Methodological novelty." Consequently, our work is now firmly positioned as a foundational catalyst for practical and scalable faithful reasoning in this domain. *(Addressing concerns from Reviewers bTxE, oaDR, xYJj)*
2.  **Validation of Faithfulness and Data Integrity.**
The core claim of faithfulness was a central point of discussion. We decisively addressed this by first clarifying that our pipeline constrains generation with dual verified inputs, including bounding boxes and overall captions, both derived directly from the established source datasets (e.g., VRSBench, DOTA), which have been rigorously verified by the community, transforming the task from open-ended invention to a constrained rationalization of known facts. To further substantiate this, we disclosed our full scientific audit protocol: a 200-sample audit conducted by three domain-expert Ph.D.s, which yielded a Fleiss' Kappa of 0.88, statistically validating our 96% alignment claim. We also provided a dataset subset for independent scrutiny. This comprehensive response was **confirmed by Reviewer xYJj** to have effectively addressed their concerns. *(Addressing concerns from Reviewers bTxE, aXVe, sTsT, xYJj)*
3. **Demonstration of Data Efficiency & Scaling.**
Concerns regarding data advantage were addressed from two perspectives.
**Firstly**, we demonstrated that RSThinker outperforms the SOTA generalist VoCoT by a massive margin (+75.8 mAP) despite using **orders of magnitude less data** (380k vs. >3M). **Secondly**, we conducted a **computationally intensive scaling study** on 50k/100k/200k subsets, confirming that our performance gains scale consistently with data, justifying the 380k corpus as an efficient and necessary scale. This evidence refutes the "data volume" hypothesis, highlighting the efficiency of our architecture. *(Addressing concerns from Reviewers sTsT, Gm5N)*
4. **Extensive Generalization Experiments.**
To validate generalization beyond the training distribution, we executed two critical tests. First, our pipeline was successfully transferred to the **smaller Qwen3-VL (2B/4B)** architecture, yielding a significant +24.5 mIoU gain, proving its model-agnostic applicability. Second, on the strictly unseen OPT-RSVG benchmark, RSThinker established a +10.18 mIoU lead over baselines, demonstrating robust zero-shot reasoning capabilities. *(Addressing concerns from Reviewers Gm5N, oaDR)*

---

> ### Author Response · Authors · 2025-11-30
> **Summary of Rebuttal Outcomes: Final Consensus Updating the Initial System Scores (2/2)**
>
> 5. **Comprehensive Contextualization with Prior Arts.**
> In response to reviewer feedback, Section 2 has been significantly expanded to discuss recent generalist frameworks (e.g., Visual CoT, VoCoT) and geographic reasoning works (GeoChain, GAEA). This revision explicitly delineates the distinction between their focus on semantic reasoning for ground-level images and our unique contribution in enabling perceptual grounding for overhead Earth Observation. *(Addressing concerns from Reviewers bTxE, oaDR, sTsT)*
> 6. **Validation against Few-Shot Prompting.**
> To test alternatives, we empirically demonstrated that few-shot prompting (k=2) leads to performance degradation across baselines in this domain, likely due to contextual interference. This result validates the necessity of our weight-based alignment approach over simpler inference-time prompting techniques. *(Addressing concerns from Reviewer sTsT)*
>
> We respectfully submit this summary to bridge the discrepancy between the frozen metadata and the substantive consensus acknowledged by the reviewers. We hope this clarified context assists in a fair and comprehensive assessment, and we truly appreciate your understanding and support in navigating this situation.

---

### Meta-Review · Area_Chair_BKQk · 2026-01-06

**Summary:**

This submission received an extensive discussion and rebuttal.

The reviewer's concerns can be summarized as below:
1. Missing / lacking discussion / comparison of some related / SoTA works (Reviewer bTxE, oaDR)
2. Methodology novelty / contribution (Reviewer bTxE, oaDR, xYJj)
3. Evaluation related concerns including the correction of the metrics, the verification of the reasoning / chain of thought. (Reviewer bTxE, Gm5N, aXVe, xYJj)
4. Request for the additional experiments for verifying the effectiveness of the approach (Reviewer bTxE, Gm5N, aXVe)

**Reviewer Concerns:**

For the above concern, i've read the rebuttal and the comments.

1. Related work: The updated submission increased the discussion about the related works. The differences b/w the proposed approach and the related works have been extensively discussed and analyzed. The concerns related to the related work shall be addressed.
2. Methodology novelty / contribution: By reading the comments / rebuttal, personally I agree with the reviewers that the proposed approach (methodology speaking) is not very novel. The main contribution of the paper is demonstrating the strong performance brought by this proposed approach (SFT + RL) Given that, I think the novelty part is not fully addressed. However the contribution should be good enough for a publication.
3. Evaluation related concerns. During the rebuttal, the author provides detailed responses to the metrics correctness part. For concerns related to reproducing the results (Gm5N), I don't think the author fully addressed that. By reading the rebuttal, the Geo-Contextualized Prompt seems still not be provided. For the verification of the chain of thought (sTsT), the author provided results and the approach for verifying the chain of thought.
4. For additional experiments, the author provided detailed and extensive responses.

**Reviewer Scores:**

For bTxE, given the rebuttal, I think the reviewer should improve the rating.
For Gm5N, given the comment 'I will certainly raise the score if authors address (some) my concerns.', I believe the criteria is meet. The reviewer should improve the rating.
For oaDR, the reviewer concerns should be addressed.

---

### Decision · Program_Chairs · 2026-01-26

Accept (Poster)